# Role of solvent-anion charge transfer in oxidative degradation of battery electrolytes

Eric R. Fadel[1,2,3], Francesco Faglioni [4], Georgy Samsonidze[2], Nicola Molinari[1,2], Boris V. Merinov[5], William A. Goddard III[5], Jeffrey C. Grossman [3], Jonathan P. Mailoa[2] & Boris Kozinsky [1,2]

Electrochemical stability windows of electrolytes largely determine the limitations of operating regimes of lithium-ion batteries, but the degradation mechanisms are difficult to characterize and poorly understood. Using computational quantum chemistry to investigate the oxidative decomposition that govern voltage stability of multi-component organic electrolytes, we find that electrolyte decomposition is a process involving the solvent and the salt anion and requires explicit treatment of their coupling. We find that the ionization potential of the solvent-anion system is often lower than that of the isolated solvent or the anion. This mutual weakening effect is explained by the formation of the anion-solvent charge-transfer complex, which we study for 16 anion-solvent combinations. This understanding of the oxidation mechanism allows the formulation of a simple predictive model that explains experimentally observed trends in the onset voltages of degradation of electrolytes near the cathode. This model opens opportunities for rapid rational design of stable electrolytes for high-energy batteries.

[1] John A. Paulson School of Engineering and Applied Sciences, Harvard University, Cambridge, MA 02138, USA. [2] Robert Bosch LLC, Research and Technology Center, Cambridge, MA 02139, USA. [3] Department of Materials Science and Engineering, Massachusetts Institute of Technology, Cambridge, MA 02139, USA. [4] Department of Chemical and Geological Sciences, University of Modena and Reggio Emilia, Via Campi 103, 41125 Modena, Italy. [5] Materials and Process Simulation Center, California Institute of Technology, Pasadena, CA 91125, USA. Correspondence and requests for materials should be addressed to B.K. (email: bkoz@seas.harvard.edu)

Lithium-ion batteries have become the most widespread electrochemical storage technology due to their high-energy density making them ideal in portable applications[1]. However, their implementation for applications requiring higher energy and power density such as car batteries remains a challenge[2]. The necessary energy density for their use in electric cars requires advances in cathode and anode materials, as well as electrolytes, in order to increase the operating voltage and capacity[1,2]. However, with an increase of the operating voltage, the commonly used organic electrolytes become unstable and new electrolyte materials are needed to increase safety and cycle life while possessing a lithium-ion conductivity of at least $10^{-2}$ S cm$^{-1}$ at room temperature, normally regarded as the threshold for technological viability[1]. The study of the voltage window in which a given electrolyte material remains stable, as well as the pathways of degradation of the electrolyte outside of this window, is particularly important to the design of improved battery materials and systems[3,4]. Electrolyte breakdown is a complex interface phenomenon that is difficult to characterize experimentally, and computational simulations can give valuable insights into the key microscopic mechanisms that are difficult to access experimentally[5].

When determining the stability window of an electrolyte, one should consider the rate of the oxidation (and reduction) reactions, which is linked to the activation free energy of the electron transfer. Computing this quantity accurately requires systematic explicit calculations of the reaction[6–8], and we have, for example, done this for the Li-PEO-TFSI system[9].

It is also important to consider the full oxidation reaction instead of computing HOMO–LUMO levels of individual species[10], with many different processes such as complex multi-step reactions with different molecules in the electrolyte, as well as surface effects, impacting the true voltage window of stability[10].

Although there are computational efforts trying to investigate oxidation and reduction effects at the electrolyte-electrode interfaces[11–13], progress is slow due to the computational complexity of such simulations. A practical approach is to investigate the intrinsic stability of the bulk liquid electrolyte, which can be used as a first filter when screening improved materials by providing an upper bound to the voltage stability of the entire system[14–16]. Furthermore, due to the lack of knowledge of the degradation pathways and oxidation mechanisms, it is common to use the vertical or adiabatic ionization potential (IP) as a useful approximation of the electrolyte stability[17–19].

In this direction, the majority of reported works focus on the decomposition of a single species of the electrolyte, the solvent or the anion (possibly within an implicit solvation approximation), which allows for simpler and faster calculations and enables high throughput screening of electrolytes[15,20].

In Li-ion battery electrolytes, it was noticed that studying isolated species does not fully capture the intricate interactions between solvents and anions, and instead the correct approach is to investigate systems comprising multiple and explicit solvents and anions[17,21–24]. The importance of explicit solvation was demonstrated for example in ionic liquids[25,26]. This approach has highlighted interesting properties of the oxidation processes in electrolytes consisting of multiple species, and, most importantly, shows a weakening of the solvents in the presence of anions, or even other solvents[5,17,27]. The observed weakening of the solvent was attributed to intermolecular reactions subsequent to the initial oxidation, such as hydrogen or fluorine transfer[21,22,28]. Experimental studies do not unanimously observe this phenomenon: while some report the dependence of the solvent oxidation[29–31] on the anion chemistry, others do not find such behavior[32,33]. Thus, degradation mechanisms remain poorly understood because of the complexity of the many possible pathways, and the limitations and cost of current computational methods.

In this work we explicitly treat the oxidation of anion-solvent complexes, and we focus on the onset of oxidation to estimate an approximate upper limit of the voltage stability window, without looking at possible subsequent degradation pathways and molecular geometry relaxation following the electron removal.

Therefore, instead of computing adiabatic IP, which approximates the oxidation potential, our approach of estimating the activation energy of the electron transfer relies on computing the distributions of vertical IP of molecular complexes sampled from molecular dynamics, to describe the stability in a statistical way. Vertical IP has been shown to be a useful indicator of oxidative stability[9,18,19], and we calculate it using the ΔSCF approach[15,18,26], by computing the energy difference between oxidized and initial states (without geometry relaxation).

We compare several semi-local and hybrid DFT functionals, such as PBE[34] and B3LYP[35], which are susceptible to self-interaction errors and spurious charge delocalization[36,37], and find that the M06-HF[38–40] functional is a suitable option to capture the charge density differences in a physically consistent way.

We then examine the vertical ionization of coupled solvent-anion system, using combinations of four anions: 4,5-dicyano-2-(trifluoromethyl)imidazolium (TDI$^-$), bis-(trifluoromethane solfonimmide) (TFSI$^-$), tetrafluoroborate (BF$_4^-$), and hexafluorophosphate (PF$_6^-$), with four solvents: dimethyl sulfoxide (DMSO), dimethoxyethane (DME), propylene carbonate (PC), and acetonitrile (ACN).

The main findings of this work are that (1) upon ionization the charge is removed either from the solvent or the anion depending on the electrolyte chemistry (2) in the case of solvent oxidation, the ionization potential of the combined solvent-anion system can be significantly lower than the IP of either the isolated solvent or the isolated anion; (3) this weakening effect is driven by electrostatic stabilization of the oxidized charge transfer complex forming between the solvent and the anion molecules.

Using a simple charge transfer model previously formulated in the context of molecular crystals[41], we provide an intuitive understanding of the oxidation of the solvent-anion pair, and a simple model to predict whether the anion or one of the solvents is fully oxidized when removing an electron from the total electrolyte system.

## Results

**Study of charge delocalization and self-interaction correction.** In this section, we point out the importance of managing the spurious charge delocalization issues present in semi-local DFT calculations of oxidized molecular systems consisting of multiple components. The key results are summarized in Table 1, which shows IPs and charge study for different functionals and different configurations. Gas phase anion and solvent IPs are reported in columns two and three. In columns 4–9 and 10–15, results are presented for systems of five identical solvents 10 Å apart and five identical anions 500 Å apart respectively, in vacuum. The last five columns present configurations of one TFSI$^-$ molecule solvated by three DME molecules. Structures obtained from optimizing the geometry with the different ab-initio methods are slightly different, but qualitatively the same. We find that only methods including 100% long-range HF exchange (LC-BLYP, M06-HF, and HF) correctly describe the removal of the electron from a single molecule in the cases of 5 anions or 5 solvents. In these cases, the IP is almost independent of the number of molecules in the calculation, as physically expected. Similarly, in the case of one TFSI$^-$ anion surrounded by DME solvents, only LC-

**Table 1 Ionization potential (eV), and charge distribution for different systems computed in vacuum with different functionals**

| Method | DME | TFSI⁻ | DME₅ | | | | | | TFSI⁻₅ | | | | | TFSI⁻+DME₃ | | | |
|---|---|---|---|---|---|---|---|---|---|---|---|---|---|---|---|---|---|
| | | | IP | S | S | S | S | S | IP | A | A | A | A | IP | A | S | S | S |
| PBE | 8.75 | 5.82 | 6.77 | | | | | | 3.60 | | | | | 4.99 | | | | |
| PBE0 | 9.34 | 6.86 | 8.06 | | | | | | 4.85 | | | | | 6.07 | | | | |
| B3LYP | 9.30 | 6.85 | 7.88 | | | | | | 4.70 | | | | | 5.91 | | | | |
| B3LYP-D3 | 9.30 | 6.85 | 7.88 | | | | | | 4.71 | | | | | 5.80 | | | | |
| M06-2X | 9.92 | 7.26 | 9.36 | | | | | | 6.22 | | | | | 6.85 | | | | |
| CAM-B3LYP | 9.73 | 6.72 | 9.16 | | | | | | 6.08 | | | | | 6.86 | | | | |
| LC-BLYP | 10.13 | 7.17 | 10.11 | | | | | | 7.08 | | | | | 7.23 | | | | |
| M06-HF | 10.24 | 7.93 | 10.20 | | | | | | 7.66 | | | | | 7.27 | | | | |
| HF | 8.82 | 6.30 | 8.80 | | | | | | 6.17 | | | | | 6.45 | | | | |

For reference, in the case of DME, the experimental IP is 9.8 eV and our calculated DLPNO-CCSD(T) IP is 9.9 eV. For TFSI⁻ our calculated DLPNO-CCSD(T) IP is 7.3 eV. These values are reported in Supplementary Table 8. Columns 2 and 3 refer to isolated DME and TFSI⁻, respectively. Columns 4–9: five identical DMEs 10 Å apart. Columns 10–15: five identical TFSI⁻ molecules 500 Å apart. Columns 16–21: one TFSI⁻ solvated by three DMEs. The fraction of an electron removed from each anion (A) or solvent (S) molecule is proportional to the intensity of blue in the corresponding column

BLYP[42,43], M06-HF, and HF correctly remove an electron from a single molecule, namely one of the DMEs. All other functionals, including some commonly used semi-local functionals, non-physically delocalize the charge, removing a fraction of an electron from all molecules. More results for this study are presented in the Supplementary Information, with $PF_6^-$ as the anion (see Supplementary Table 5), or showing that the results persist when adding implicit solvation (see Supplementary Table 4). Finally, we note that in the case of TFSI⁻ and three DME, the functionals that completely oxidize one molecule oxidize the solvent, which has a higher IP than the anion. This is counter-intuitive, and we discuss in detail in the rest of this work how charge transfer pair formation is the cause of this oxidation mechanism. The phenomenon of charge transfer upon oxidation of the anion-solvent complex, i.e., removal of the electron from the molecule with higher IP, is the focus of the next sections. As mentioned in the introduction, this effect was described in previous works as a consequence of reaction that follows oxidation, but here we emphasize that it occurs already in the calculations of the vertical IP, independent of chemical degradation pathways.

In summary, only functionals with 100% of HF exchange at long-range (HF, M06-HF, LC-BLYP) yield the correct ionization behaviors and do not suffer from charge delocalization, while methods without full HF exchange misrepresent the oxidized state, predicting charges delocalized on more than one molecule. We also mention that recent work on ionic liquids showed that range-separated functionals suffer much less from self-interaction delocalization error and show similar dipole moments and interaction energies as wave-function methods[37]. From our study of charge delocalization in DFT functionals, M06-HF is inferred to be an appropriate functional to study the effect of explicit solvation on ionization in a wider set of chemistries and geometries. However, we also observe that the isolated IP from M06-HF for the molecules, whether solvents or anions, are found to be overestimates of the experimental values. The inaccuracy of M06-HF is a known problem, and it persists when computing IP from optimized geometries. In this work, emphasis is placed on the correct treatment of ionization and minimizing delocalization error, focusing on the physical mechanism of ionization and the origins of charge transfer in solvent-anion complexes. Most trends presented here are significant in comparison with the IP errors and are fundamentally not altered by the inaccuracies arising from the functional. Here we also note that none of these functionals is as accurate as CCSD(T) for IP. However, their lower computational cost allows us to study systems of multiple molecules otherwise practically impossible with CCSD(T) level of theory. From the results of our DLPNO-CCSD(T) calculations

reported in the Supplementary Information (Supplementary Table 8), we find that it is possible to get very accurate IP values (compared to experimental measurements) with a more accurate method when required. Furthermore, we find that M06-HF accuracy is satisfying for the chemistries studied in this work, with an error smaller than the spread of the IP over different configurations. Furthermore, we discuss later in this paper that although the absolute values of the IP for the different systems may be different between methods, the trends and phenomena described in this work remain valid even when analyzing DLPNO-CCSD(T) results.

**IP values of anion-solvent pairs.** In this section, we present the results for the IP study of anion-solvent pairs, and provide a simple empirical formula for the IP of the pair, before proposing a simple physical model in the next section. First, we find that the spread of the IP values over all the snapshot configurations is significant (on the order of 1 eV), across all chemistries.

To investigate this spread we study two specific couples, (TFSI⁻, PC) and ($PF_6^-$, DME), using 200 configurations of the anion-solvent pair. Figure 1 shows the obtained distributions of vertical IP. The first pair comprises TFSI⁻ that is a "weak" anion (i.e., has an IP in gas phase of 7.23 eV), and PC that is a "strong" solvent (i.e., has an IP of 12.66 eV). The second pair comprises a "strong" anion ($PF_6^-$ with an average IP of 9.7 eV), and a "weak" solvent (DME with an average IP of 10.2 eV). We indeed find that the spread is significant, but that the average distribution converges for a moderate number of configurations (for (TFSI⁻, PC) for example, the average with 30 random configurations is 7.5 eV and the average with 200 random configurations is 7.45 eV). The use of 30 random configurations is deemed a good compromise between computational cost and accuracy, and therefore applied to the study of all anion-solvent pairs.

We also plot the heat maps of the initial and final energies with respect to the ionization potential (Fig. 1) for these two anion-solvent pairs. We find that there is a distinction between anion-solvent pairs for which the anion is oxidized, and those for which the solvent is oxidized. For the (TFSI⁻, PC) pair, the anion is oxidized. In this case, for configurations with IP near the average value, the configuration's initial and final energies are near the average of their respective distributions. For configurations corresponding to the lower end of the IP distribution, their initial energy is high relative to the set of all configurations and their final energy is low. Inversely, configurations with high IP have below average initial energy but high final energies. Considering the ($PF_6^-$, DME) pair in which the solvent is

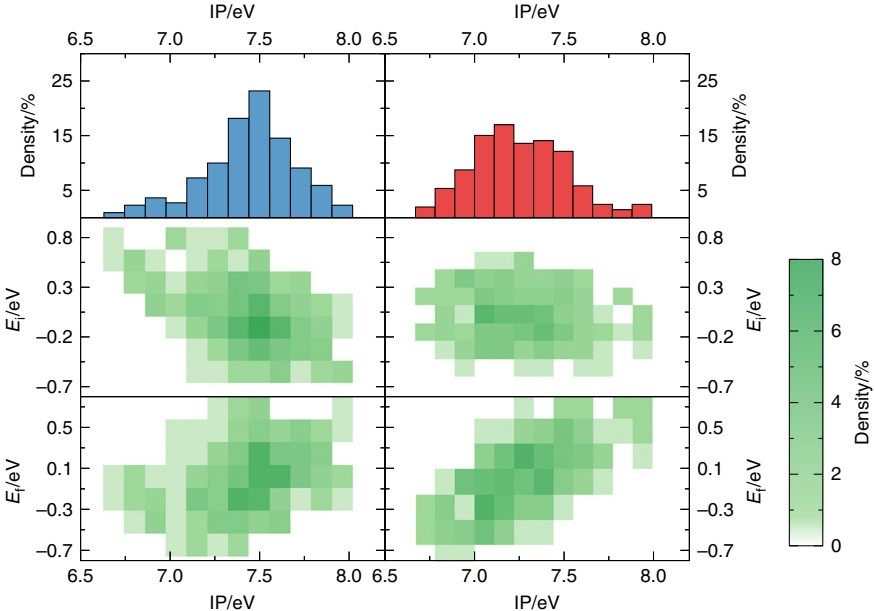

**Fig. 1** IP distribution for 200 random configurations of anion-solvent pair. The first column corresponds to the (TFSI⁻, PC) pair, and the second column corresponds to the ($PF_6^-$, DME) pair. Supplementary Information reports a full study for all anion-solvent pairs (Supplementary Fig. 3) with 30 configurations. The color code, used consistently throughout this work, represents in blue configurations where the anion is fully oxidized (all configurations of the (TFSI⁻, PC) pair) and in red configurations where the solvent is oxidized (all configurations of the ($PF_6^-$, DME) pair). Below these two figures, we plot the heat maps of the initial (middle plot) and final (bottom plot) energies for the two pairs. The energies are plotted relative to the average energy (either initial or final)

oxidized, we find a very different behavior: the initial energy is relatively uncorrelated to the IP of the configuration, and the distribution of IP is mostly governed by the final energy. This difference is significant and highlights a difference in the oxidation mechanism which can be understood using the model discussed below.

Because of this spread, we conclude that sampling many different configurations is essential to properly describe oxidation of realistic solutions at finite temperature. We also note that in principle the stability of the system could be inferred from the IP distribution across different solvation structure configurations. Neglecting interface reactions and considering only intrinsic oxidation stability, one can suppose that the onset of electrolytes' degradation is determined by the lower edge of the distribution of IP values. Indeed, assuming that electron transfer during oxidation occurs much faster than the nuclear dynamics of the system, and given that many different configurations will be explored over time in the vicinity of the electrode, those configurations that are more easily oxidized will limit the stability of the electrolyte. The whole IP distribution allows to understand the stability of the electrolyte as the voltage is increased past this threshold value. Indeed, in cyclic and differential voltammetry experiments the measured electrical current due to degradation exhibits gradual increase with electrode voltage, qualitatively consistent with our computational result. For all the chemistries we studied, the IP distribution has a single peak and is centered at its average value (see Supplementary Fig. 3). Interestingly, the spread of the distribution was not found to depend significantly on chemistry, but does depend on which species (the anion or the solvent) is oxidized during overall oxidation (this is discussed in detail below and in the Supplementary Information). In the rest of the study, we therefore focus on IP trends inferred from the average IP across several configurations as these averages accurately represent the IP distributions.

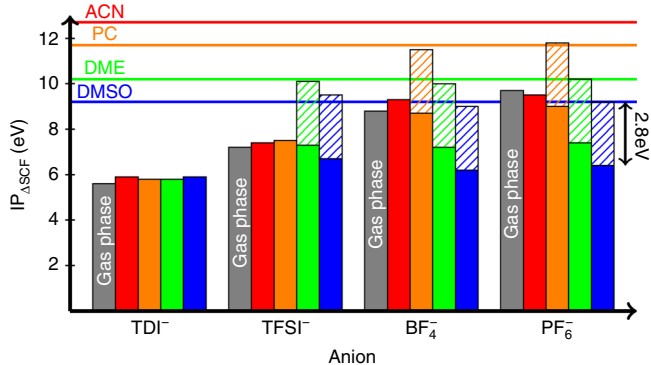

**Fig. 2** IP of pairs of one anion and one solvent molecule with respect to the isolated species. Bars represent the average over 30 configurations of each pair, with the x-axis showing the anion and the color indicating the solvent used (blue: DMSO; green: DME; orange: PC; red: ACN; gray: isolated anion). Values for the isolated species (y-axis for the solvent, gray bar for the anion) are averaged over 50 configurations. Dashed bars represent constant shifts $\delta = 2.8$ eV (drawn for pairs where the solvent is oxidized). IPs are well approximated by Eq. (1): the smaller of anion IP or the solvent IP minus $\delta$

The average IP of 30 configurations for each anion-solvent pair are reported in Fig. 2. The IP values of the anion-solvent exhibit a non trivial relationship to the IP values of each individual component. Contrary to common assumptions, the overall IP is not determined by that of the solvent alone, as shown by the lack of solvent dependence in the IP of combinations containing TDI⁻ (weakest anion). On the other hand, the solvent has a clear effect on the IP of combinations containing $PF_6^-$ (strongest anion). Importantly, in no case is the couple IP equal to that of the

solvent. We now examine this trend quantitatively and later will provide a physical explanation. This overall trend is captured quite well by Eq. (1), where $A^-$ is the anion, S the solvent, $[AS]^-$ is the pair of anion and solvent before oxidation and $\delta = 2.8$ eV:

$$\text{IP}_{\text{fit}}([AS]^-) = \min \begin{cases} \text{IP}(A^-) \\ \text{IP}(S) - \delta \end{cases} \quad (1)$$

When the IP of the isolated anion is smaller than that of the solvent by more than $\delta$, for example, in the case of pairs (TDI$^-$, DMSO), (TDI$^-$, DME), or (TFSI$^-$, ACN), the combined system IP is roughly that of the anion, and the charge study shows that the electron is removed from the anion. On the other hand, when the IP of the isolated anion is greater than the IP of the solvent minus $\delta$, for example, for the pairs (BF$_4^-$, DMSO), (BF$_4^-$, DME), or (PF$_6^-$, PC), then the combined system is weaker than either species, and the IP of the pair is approximately equal to the IP of the isolated solvent minus $\delta$. Our charge study in this case shows that the solvent is oxidized, which is contrary to common intuition, given that the solvent always has higher IP than the anion. This is highlighted in the graph using striped bars of height $\delta$ showing the difference between the isolated solvent IP and that of the anion-solvent pair. This behavior corresponds to switching from oxidizing the anion to oxidizing the solvent, in which case the IP becomes independent from the anion and equal to the solvent IP minus $\delta$. This indicates that for many solvent-anion combinations a charge transfer complex spontaneously forms upon oxidation, depending on the IP of individual species, with an electrostatic stabilization $\delta$. We note that our combined simulations properly oxidize one molecule only (whether the anion or the solvent), further validating our choice of the exchange-correlation functional. Naturally, in borderline cases where the values in Eq. (1) are similar, oxidation of either the anion or the solvent may be observed, depending on the specific geometry. In this article, we refer to the IP computed using Eq. (1) as IP$_{\text{fit}}$.

We examined this effect using more accurate DLPNO-CCSD (T) calculations for the (TDI$^-$, PC), (TFSI$^-$, PC), (BF$_4^-$, PC), and (PF$_6^-$, PC) combinations, and the full results are reported in the Supplementary Information (Supplementary Tables 8 and 9). In brief, we find not only that using a highly accurate level of theory yields IP values that closely agree with experimental data, but also that the trends presented in this work remain unaltered (see Supplementary Table 9). This validates our computational approach with regards to the choice of the DFT functional.

**Interpretation of previous experiments and computations**. For further supporting evidence to our hypothesis, we compare our results with previous experimental and computational works from the literature. Comparison between computed IP and experimental IP as measured in gas phase is provided in the Supplementary Information. Focusing on the trends of oxidation voltage with respect to the choice of anion-solvent chemistry, we argue that the observed behavior will depend on which species is oxidized. When considering electrolytes with the same solvent species and varying anion species, the oxidation voltage will increase with increasingly "strong" anion (while the species that is oxidized is the anion, i.e. it is "weak" compared to the solvent), until it saturates when switching happens and the solvent becomes the oxidized species. Therefore, we expect a dependence of the oxidation voltage on the anion, but only for anions that are "weak" enough. When keeping the same anion and changing the solvent species, two different effects are at play to determine the oxidation voltage: the oxidation mechanism (whether the anion or the solvent is

oxidized) and the solvation effect, which leads to a more ambiguous, chemistry-dependent trend. Looking at the experimental observations by Ue et al.[30,44], they first find that for a given solvent, (PC), the combined system oxidation potential depends on the anion IP, but that this dependence saturates for anions such as BF$_4^-$ or stronger. Indeed, in their experimental results, the oxidation voltage for BF$_4^-$, PF$_6^-$, and AsF$_6^-$ are roughly the same, which in our understanding of oxidation, hints at a switching from anion oxidation to solvent oxidation. Then, they use a more oxidation-resistant solvent, glutaronitrile, in order to determine the anodic stability order of those anions. Thus, we find a situation where both anion and solvent oxidation matter, depending on which species is oxidized, and their study shows the two possible behaviors of oxidation voltage with changing anion. All these observations are consistent and can be explained with our new understanding of oxidation. However, direct comparison to oxidation voltages as obtained through cyclic voltammetry measurements is not a well-defined validation procedure for computational methods. These measurements have a large variance as they are affected by a wide range of parameters such as the scanning rate[30], the nature of the electrode, the concentration of species in the electrolyte, and even the method adopted to infer the oxidation voltage from the raw data[45]. On the other hand, computational method suffer from DFT functional and basis set inaccuracies and approximate solvation effects. In this work, emphasis was given to accurately describe oxidation mechanisms from the microscopic, ab-initio standpoint, compromising on the accuracy on the absolute value of the IP (also, surface effects are not taken into account). Furthermore, we did not take into account zero-point energies and vibrational entropy. Whilst, for these reasons, quantitative comparison with experimental results is difficult, we believe that the trends presented in this work give important insights into the possible oxidation scenarios. In the case of (TFSI$^-$, DMSO), for which detailed experimental data are available[3], we obtain an excellent agreement with the observed oxidation potential, as reported in a separate publication[9].

Previous computational work by Kim et al.[27] presented, without elaborating, the location of the HOMO for different anion-solvent pairs, finding that the HOMO of the pair can be on the anion or on the solvent. Even though the exchange-correlation functional used (M06-2X) still has a degree of spurious delocalization (Table 1), the trend in the reported data is consistent with our findings. In the work by Kim et al.[27], the HOMO is always on the anion for a very weak anion (bis(oxalate) borate) across all studied solvents. However, for a strong anion (PF$_6^-$), the HOMO is always on the solvent across all studied solvents. In intermediate cases such as for TFSI$^-$, the HOMO is on the solvent for the weaker solvents, but is on the anion for stronger solvents. The decrease of the overall IP of the anion-solvent combination relative to the IP of each species is explained in that work, and earlier ones[17,46], as a consequence of intermolecular chemical reactions. Thus, we see strong evidence emerging that the oxidation of the electrolyte is controlled by both the anion and the solvent. However, in contrast to previous interpretations, we conclude that the weakening effect is driven by the spontaneous charge transfer between the anion and the solvent and does not require consideration of the specific reaction pathway following oxidation.

**Charge transfer stability model**. To understand the full chemistry dependence of electrolyte stability in terms of the solvent-anion charge transfer complex formation, we derive a simple stability model, similar to the one used in the field of charge transfer in

molecular crystals[41]. For a pair of molecules of anion $A^-$ and solvent S, we can express the energy of the oxidized pair as:

$$E\left([AS]^0\right) = \min\begin{cases} E(A^0 S^0) \\ E(A^- S^+) \end{cases} \quad (2)$$

Therefore, there is a trade-off between oxidizing the anion which is weaker or oxidizing the solvent and creating a dipole which lowers the electrostatic energy. This formula accounts for the trend presented in the previous section: when the anion is significantly weaker, the first expression is the minimum, i.e., the electrostatic gain is not sufficient for the solvent be oxidized. In cases where the anion IP is not significantly lower than the solvent IP, it will be more energetically favorable to form an electrostatic dipole by oxidizing the solvent. The threshold between the two cases is determined by the electrostatic energy $\delta$ of the dipole formed by oxidizing the solvent, as well as the individual IPs.

By considering the total energy of the system as a sum of the short-range quantum contribution (ionization) and a long-range electrostatic contribution (dipole energy), it is possible to derive from Eq. (2) an expression for the pair IP. Denoting the classical electrostatic energy $E_e(X)$ for the molecular system $X$ considered in isolation, and $E^{bind}(XY) = E_e(XY) - E_e(X) - E_e(Y)$, we have:

$$IP_{model}\left([A^- S^0]\right) = \min\begin{cases} IP(A^-) + E^{bind}([A^0 S^0]) - E^{bind}([A^- S^0]) \\ IP(S^0) + E^{bind}([A^- S^+]) - E^{bind}([A^- S^0]) \end{cases}$$
$$(3)$$

The only approximation used to derive this expression is that the quantum (non-electrostatic) contribution to the total energy of a solvent-anion pair is short-ranged and is close to the sum of each component's quantum energy contribution. At the same time, the long-range electrostatic energy can be treated classically in each case. This allows decoupling the electrostatic and the ionization contributions (see Supplementary Information for the detailed derivation). The IP as computed using Eq. (3) is called $IP_{model}$ in the rest of this work. From this formula, we can identify the empirical value of $\delta$ from $IP_{fit}$ in Eq. (1) as the electrostatic dipole energy ($E_e([A^- S^+]) - E_e(A^-) - E_e(S^+)$). The fact that this value seems constant across the different chemistries is due to the fact that the electrostatic energy depends on the anion-solvent distance and their unit charges, and for all the systems reported here this distance is roughly the same considering averages across different configurations.

In order to examine the accuracy of this model, we use the same geometries for the isolated anion and solvent to compute their vertical IP (without changing the geometry). The electrostatic energies are computed for the isolated anion and solvent, for the initial and oxidized cases, as well as for the pair combinations, and from that we obtain the electrostatic contributions appearing in Eq. (3). Electrostatic energies are defined and computed as the sum of the core–core interactions, the core–electron interactions, and the Hartree energy for the classical electron–electron interaction (all of which are extracted from the DFT computations for the isolated species as well as pairs). We see good agreement between $IP_{\Delta SCF}$ and $IP_{model}$ across most configurations. Thus, the description of the charge transfer effect using decoupling of the quantum and electrostatic energies of the isolated molecules and pairs seems to hold up across multiple chemistries and configurations. The most significant deviation of the model from the $\Delta SCF$ result occurs for configurations involving PC and strong anions, (BF$_4^-$, PC) and (PF$_6^-$, PC). The only approximation in the model is the absence of coupling in quantum energy between the anion and solvent, therefore, these deviations are possibly due to anion-solvent

**Table 2 Ionization potentials for 16 anion-solvent pairs computed with different approaches**

| | Anions ⇒ | TDI$^-$ | TFSI$^-$ | BF$_4^-$ | PF$_6^-$ |
|---|---|---|---|---|---|
| ⇓ Solvents | IP$_{\Delta SCF}$ IP$_{model}$ IP$_{fit}$ S %, A % | 5.63 | 7.23 | 8.79 | 9.70 |
| DMSO | 9.20 | 5.85 5.93 5.63 0%, 100% | 6.66 6.60 6.40 100%, 0% | 6.24 6.27 6.40 100%, 0% | 6.41 6.37 6.40 100%, 0% |
| DME | 10.24 | 5.76 5.88 5.63 0%, 100% | 7.28 7.09 7.23 10%, 90% | 7.20 7.16 7.44 100%, 0% | 7.44 7.14 7.44 100%, 0% |
| PC | 11.72 | 5.90 6.06 5.63 0%, 100% | 7.52 7.26 7.23 0%, 100% | 8.75 9.73 8.79 100%, 0% | 9.00 10.39 8.92 100%, 0% |
| ACN | 12.66 | 5.77 5.79 5.63 0%, 100% | 7.38 6.78 7.23 0%, 100% | 9.28 9.61 8.79 60%, 40% | 9.54 9.63 9.70 94%, 6% |

coupling, given that these anions are compact and may approach the solvent closely enough. We note, however, that analysis of charge densities confirms that only one molecule is oxidized in all cases, meaning that charge transfer is complete. Table 2 summarizes the main findings of this work, showing the average $IP_{\Delta SCF}$, the average $IP_{model}$ as predicted from the simple charge transfer model (Eq. (3)), $IP_{fit}$ (Eq. (1)), as well as the species that is oxidized. We see that these last two models predict quite well the full first-principles $IP_{\Delta SCF}$ of the anion-solvent pairs. This provides clear evidence that the anion-solvent weakening effect observed in our $\Delta SCF$ calculations originates from charge transfer and electrostatic coupling between the two species. We can also explain why the spread of the IP distribution is smaller for chemistries where the anion is oxidized than for chemistries where the solvent is oxidized. Indeed because the dipole interaction energy $\delta$ depends on the relative distance and orientation of the solvent around the anion, this energy - and therefore the IP of the pair - varies more between different configurations. Thus for two different chemistries where the average IP of the pair is roughly the same, the onset of the electrolyte degradation may happen at lower voltages if the solvent is the oxidized species, due to a wider spread of the IP values.

Finally, we further demonstrate the validity of the charge transfer model for these systems by studying a case where all oxidations lead to dipole formation, and show that in such cases, the IP increases with distance until the dipole energy is too low to energetically favor oxidation of the solvent. At this point the anion is oxidized and the charge transfer effect disappears. Increasing the distance past this point does not change the total IP (which is expected since neither the initial nor the final state involves intermolecular electrostatic interaction to first order, unlike the dipole case). This is reported in Fig. 3. The leftmost point in the plot (shortest distance) corresponds to the configuration as taken from the MD snapshot, without increasing the intermolecular distance. Note the two regions, one where it is energetically favorable to oxidize the solvent (left part) and the right part where it is more favorable to oxidize the anion. The

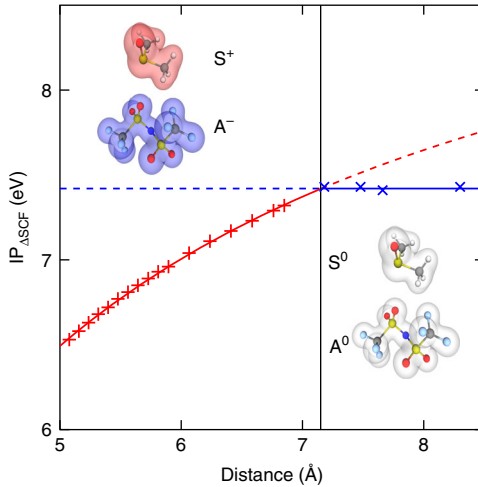

**Fig. 3** $IP_{\Delta SCF}$ for a (TFSI$^-$, DMSO) configuration, with respect to the increasing distance (leftmost point in the plot is the initial configuration). The red marks correspond to situations where the solvent is oxidized, the blue marks to situations where the anion is oxidized. The distance plotted here is the distance between the nitrogen atom of the TFSI$^-$ anion and the sulfur atom of the DMSO solvent. The IP for the right part of the plot (where the anion is oxidized) is constant with distance. As shown by the solid red line, the dependence of the left part of the plot is accurately captured by a fit with the formula $IP = \text{constant} - \frac{\alpha}{r}$

increase in IP in the left part fits the following formula:

$$IP\big([A^- S^0]\big) = IP(S^0) - \delta \approx IP(S^0) - \frac{\alpha}{r} \qquad (4)$$

where the second term represents the dipole energy of point charges. The coefficient $\alpha$ is found to be 15 eV Å, which is close to 14.4 eV Å, corresponding to the case of two point unit charges in vacuum. The distance $r$ considered here is taken to be the distance between the nitrogen atom of the TFSI$^-$ anion and the sulfur atom of the DMSO solvent. We verify that the IP in the right part of the plot is constant with respect to distance. Thus even a simple point charge electrostatic model accurately captures the charge transfer transition and can therefore be used to estimate the contribution of intermolecular electrostatics to the balance between anion or solvent oxidation. We also note that the overall IP is a non-decreasing function of the separation distance, which suggests that larger molecular species may be more favorable for stability, keeping IP values constant. This consideration points to the possibility of optimizing the electrolyte's stability not only by varying the IP of the anion and the solvent but also by tuning the anion's first solvation shell radius.

**Possible degradation mechanisms.** Previous studies examined the possibility of hydrogen transfer after oxidation[17,22,27], suggesting that it is the reason for the weakening of the combined solvent-anion system. We have shown that the weakening effect of the anion-solvent pair can be explained regardless of any specific degradation steps following the system oxidation. However, the study of electrostatic intermolecular interactions shown above can give new insight into the oxidation-driven reaction mechanisms. Without doing an exhaustive study of reaction mechanisms and their energy barriers, this section focuses on the impact of the proton (ionic charge) transfer mechanism. Here we note that this study is done in vacuum, whereas the true degradation mechanism involves the coupling of different processes including solvent reorganization and molecular relaxation after the electron removal. Thus, this study only provides an example

of possible evolution following the charge transfer complex formation. We postulate that H transfer is energetically favorable in the cases of charge transfer complexes partly because of electrostatics, since it would compensate the dipole formation and lower the electrostatic energy. In this work, using the same configuration of anion-solvent pairs, no spontaneous intermolecular reaction was observed when we relaxed the geometries. We proceeded to study H transfer by initially displacing H towards the anion, followed by relaxation of the oxidized structure. We found that in those anion-solvent pairs where the charge transfer dipole was formed (i.e. solvent oxidized), an H atom from the solvent was observed to transfer to the anion in about 80% of the configurations. In all cases, if the structure was not oxidized, the H atom relaxed to the initial structure. For the anion-solvent pairs where oxidation results in electronic charge transfer, hydrogen transfer indeed lowers the dipole moment and total energy of the system. In the case of BF$_4^-$ and PF$_6^-$ anions, the hydrogen atom transfers to a fluorine, forming HF, leaving a BF$_3$ or PF$_5$. Figure 4 shows typical snapshots of configurations with charge transfer. We conclude that hydrogen transfer is not the cause of the electrolyte weakening but rather a consequence of the intrinsic electronic charge transfer complex formation, governed by the interplay between (quantum) ionization and (classical) electrostatic dipole energetics. A detailed research of degradation mechanisms and energy barriers in light of the findings of this work was performed for (TFSI$^-$, DME)[9]. Other combinations will be addressed in a future article.

**Effects of solvation.** So far we explicitly considered pairs of anion and solvent molecules in vacuum. In this section, we examine the effect of solvation on the oxidation energetics and the electrostatic interaction between the electrolyte species. Our main finding is that solvation quantitatively changes the electrostatic dipole energy in the presence of solvent (denoted by $\delta_{\bullet}$) primarily due to the dielectric screening effect due to the solvent, and we explain the trends across several solvent-salt combinations again using a simple electrostatic model derived from the above understanding of the charge transfer complex. First we look at the dependence of the IP on the number of solvent molecules in the explicitly solvated scenario. We find that the average IP increases with the number of solvents (up to five solvent molecules, see Supplementary Fig. 5). This increasing trend is expected from classical electrostatic energy of a charge in a dielectric medium, and can be understood as a polarization effect of the additional solvents, i.e., that anion (negative species) IP increases and solvent (neutral species) IP decreases with the solvent dielectric constant. We note in passing that in order to obtain accurate IP values for explicitly solvated systems, a proper extrapolation to large system size is needed[47], which requires expensive simulations that lie outside the scope of our investigation. It is also important to note that in all the explicitly solvated computations there is still only one species that is fully oxidized upon removal of charge (whether it is the anion or one of the solvents). We again emphasize that for this to happen it is critical to choose an exchange-correlation functional with minimal delocalization errors, such as M06-HF. Therefore, the smallest unit that is needed to study oxidation is the explicit anion-solvent pair, and addition of solvation effects the results only through polarization. To analyze the long-range effect of solvation and estimate $\delta_{\bullet}$, we build on the previous finding with explicit solvents that only one molecule is oxidized and assume that the electron removal is much faster than any other process, obtaining Supplementary Eq. (4). We employ the PCM implicit solvent model for all the BF$_4^-$ pairs (i.e., BF$_4^-$ solvated with DMSO, DME, PC, or ACN) to estimate $\delta_{\bullet}$. In order to compute the effect of PCM implicit solvation on the vertical IP,

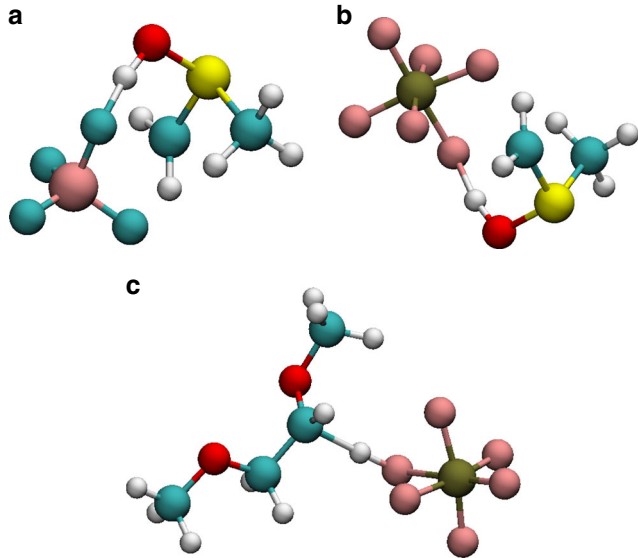

**Fig. 4** Hydrogen transfer for different anion-solvent pairs. **a** ($BF_4^-$, DMSO), **b** ($PF_6^-$, DMSO), **c** ($PF_6^-$, DME)

one must account only for the optical screening effect of solvation, which we do with the method provided in the Gaussian code[48,49]. We use the following static and high-frequency dielectric constants ($\epsilon_0$, $\epsilon_\infty$) of (46.8, 4.16) for DMSO, (4.24, 2.16) for DME, (65.5, 4.14) for PC, and (35.7, 4.0) for ACN[50,51]. We find that the solvent is still oxidized, like in vacuum, and the value of the difference $\delta_\bullet$ between the IP of the solvent and that of the solvent-anion pair is indeed lower but still significant in the PCM-solvated calculations. The values of $\delta_\bullet$ are reported in Supplementary Table 7. Thus we find that implicit solvation does not change the qualitative picture seen in the vacuum case. As discussed above, after the electron removal, it is very difficult to assess how the solvent reorganization and other subsequent reactions are affected by the nature of the oxidized state.

## Discussion

This study shows that oxidative stability of Li-ion battery electrolytes is governed by non trivial coupling between anion and solvent, and requires their coupling to be simulated explicitly. We find that only one molecule, either the solvent or the anion, loses an electron upon oxidation, but the value of the ionization potential (IP) depends on the chemistry of the components. The overall oxidative stability of the combined solvent-anion system is often significantly lower than the stability of each individual species, and increasing the IP of one of them does not necessarily increase the stability of the resulting electrolyte. By computationally examining a wide range of anion and solvent combinations we find a universal coupling behavior which is explained by the formation of a charge transfer complex upon oxidation, depending on the IP of anions and solvents and their electrostatic interaction. We construct a simple model based on this understanding that is able to quantitatively capture the counterintuitive trends observed in ΔSCF ionization potentials and predicts trends that are consistent with experimental observations. We emphasize that common semi-local density functionals suffer from charge delocalization errors when describing oxidation of representative molecular clusters and are likely to miss the qualitative features and the magnitude of the charge transfer effect that is determined by the electrostatic interaction between local charges resulting from ionization. Using this model, we show how the IP of the pair can be approximated in a simple way. We find that the resulting final state of the electron removal may impact the decomposition

process, however, more investigation is needed to understand the coupled effects of solvent reorganization, molecular relaxation and how much the charge transfer may impact the subsequent decomposition reaction. In vacuum, we show how the dipole formation may facilitate Hydrogen abstraction as a subsequent step in solvent decomposition. Results presented here provide direct implications and quantitative rules for designing stable battery electrolytes, emphasizing that both solvent and salt anions must be optimized as a whole.

## Methods

**Study of charge delocalization and self-interaction correction.** Throughout this work, the vertical IP is computed using the ΔSCF formalism[15,18,26], denoted with $IP_{\Delta SCF}$. For each selected geometry two SCF computations are performed, one for the closed shell reduced state and one for the spin-unrestricted open shell oxidized state. $IP_{\Delta SCF}$ equals the difference between the energies obtained for the two states. The vibrational contribution to the energy is neglected, as it is commonly found to account for a small correction to the IP[19,20,22,52]. All the reported partial charges are estimated using the Mulliken charge scheme[53], and summed over every molecule. The study of the effect of delocalization errors in different DFT functionals for systems of multiple TFSI⁻ anions and/or DME solvents is performed using the Gaussian09 software[48], with the 6-311++G**[54,55] basis set on all atoms. For each computational method considered, we computed vertical IPs at the optimized geometry for the reduced state.

**IP values of anion-solvent pairs study.** In the second part of this study, we look at anions solvated by a different number of solvent molecules. Previous works have shown the importance of sampling electrolyte configurations[18,19,23,25], and have done so using classical MD[18,19] or ab-initio MD[25]. We note here that in our investigations we do not report results for complexes involving Li⁺ cations, since these always have higher IP, as previously reported[22], confirmed by our calculations and expected from electrostatic considerations. Thus, the most relevant configurations for oxidation, i.e., with the lowest IP, are those where the cation is not present. In this work, the structures are obtained from snapshots of classical MD simulations, obtained as follows.

First, the anion of interest and the solvent molecules are placed on the vertices of a three-dimensional cubic grid, with the aim to create a low-density non-overlapping initial structure[56,57]. Once generated, the structures are brought close to equilibrium by a series of energy minimization, compression/decompression, and annealing stages, broadly based on previous works[58,59], to overcome local energy barriers in search of lower energy minima, and, ultimately, more representative structures. The structures are then evolved using the velocity-Verlet algorithm, with a time step of 1.0 fs, in the constant number of atoms, pressure, and temperature (or NPT) ensemble. Temperature and pressure are kept at 300 K and 1 atmosphere, respectively, with a Nosé-Hoover barostat and thermostat[60–62]. The coordinates of all atoms are saved every $10^3$ timesteps (i.e., 1 ps) to ensure sufficiently uncorrelated structures. In post processing the positions of the anion and of the closest X solvent molecules (X ranging from one to five) are extracted from the snapshots and used in the ab-initio calculations. The structures do not undergo any further geometry optimization, instead we sample different configurations of this system in order to gather statistics on the IP of anion and solvent complexes. All molecular dynamics simulations are performed in the LAMMPS simulation package[63]. The interactions are modeled using the OPLS2005 force-field[64] from Schrodinger Inc. Since the MD structures do not undergo further geometry optimization, it is important to ensure that the configurations are well sampled by the classical energy model approximation. To this end, we consider a set of 200 configurations of the (TFSI⁻, PC) pair and plot in Fig. 5a the energy for these configurations computed with the classical force-field (red) and with DFT (blue). The two distributions are Gaussian-like, with the classical force-field underestimating the energy of the configurations compared to DFT, but preserving the distribution of the energy in the phase space. Thus we conclude the OPLS2005 force-field samples configurations with reasonable accuracy. Figure 5b shows an example of a snapshot for the solvated anion (for clarity, not all solvent molecules are shown), in the case of the (TFSI⁻, DME) pair, and Fig. 5c–e examples of extracted configurations with one anion and its closest solvent for the same pair.

Ab-initio calculations were performed using the M06-HF hybrid functional with the NWCHEM software[65], and for each pair we computed the IP of 30 different configurations obtained from the MD run. All calculations were spin-unrestricted, and the spin contamination of the system was consistently checked. The basis set used for all atoms was aug-cc-PVTZ[66]. No implicit solvation model was used for this part of the work, because as we wish to study the IP of explicitly solvated anions with a functional that does not induce erroneous charge delocalization. The effect of implicit solvation is also studied, and reported in the last section of the Results section. Finally, coupled-cluster with single, double, and perturbative triple corrections (CCSD(T)) calculations were performed to validate our approach. These calculations were performed using ORCA[67], in the Domain-Based Local Pair-Natural Orbital Coupled Cluster (DLPNO-CCSD(T))

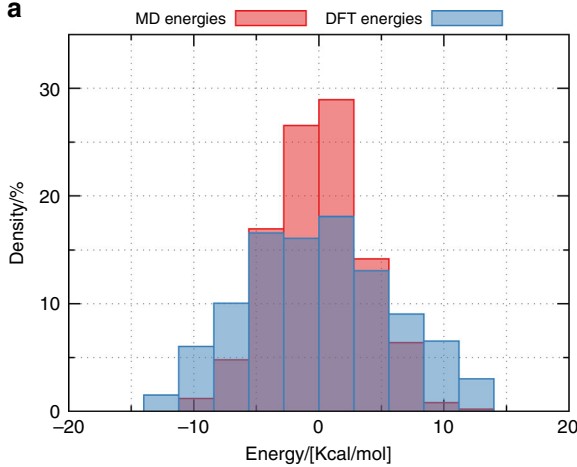

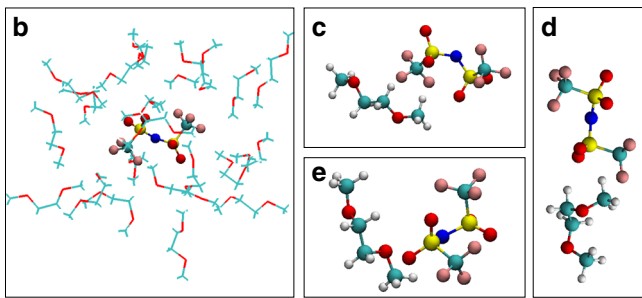

**Fig. 5** Figure **a** represents for 200 configurations of anion-solvent pairs from MD snapshots for the (TFSI⁻, PC) pair, the distribution in energy computed from MD and the distribution of energy computed from DFT (M06-HF). Both these distributions are plotted relative to their average. **b** shows a solvated anion for an MD snapshot in the case of the (TFSI⁻, DME) pair. **c**–**e** show examples of configurations of solvent-anion pair from MD snapshots for the (TFSI⁻, DME) pair

approximation. The basis set for these calculations was aug-cc-PVDZ[66]. For reference, we also compute the IP of the isolated species (anion or solvent in vacuum). The IP shown for the isolated species is the average IP from 50 different configurations taken from MD (created with a single molecule in a cubic box with 100 Å side and using the same force field).

## Data availability

The datasets generated during and/or analysed during the current study are available in the figshare repository, with the identifier [https://doi.org/10.6084/m9.figshare.8162132.v1].

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

## Acknowledgements

We acknowledge useful discussions with Heather Kulik and Mordechai Kornbluth. Funding for E.R.F. was provided by Robert Bosch LLC, partly through the MIT Energy Initiative fellowship.

## Author contributions

E.R.F., G.S., F.F., and B.K. performed the quantum calculations, formulated, and verified the charge transfer model. N.M. and J.P.M. performed MD calculations and extracted the structures. W.A.G., B.V.M., and J.C.G. gave technical support and conceptual advice. B.K. conceived and supervised the study.

## Additional information

**Competing interests:** The authors declare no competing interests.

