## [Peer Review File · Nature Communications]

Editorial Note: Parts of this peer review file have been redacted as indicated to remove third-party material where no permission to publish could be obtained.

Reviewers' comments:

Reviewer #1 (Remarks to the Author):

The authors apply DFT and other computational techniques to study issues related to oxidative stability of solvent molecules and salt (anions). They compute vertical ionization potentials (IP) and find that cooperative effects, i.e., charge-transfer between solvent and salt, may lower the IP.

My main concern is that this paper is mostly of interest to the computational community. If the paper was aimed at the battery community, I believe it has failed to stake claim to much relevance. I don't believe there is a single comparison with experimental values. The main problem is that the authors focus on vertical IP, which is generally not what is measured. The experimentally relevant quantity is clearly the oxidation potential, which is related to the adiabatic (atom-relaxed) IP. Having read this paper, I am unable to come to a conclusion regarding whether any of the issues discussed are important for battery operations. (If the authors had listed the predicted oxidation potentials, the reader could have look at whether the issues raised are relevant by looking at whether the predicted oxidation potentials fall in the range relevant to cathode oxide materials.)

So, as a paper about batteries, I definitely recommend against publishing this work in Nature Communications. Incidentally, the authors should cite works by the Borodin group about the effects of different molecular configurations on oxidation, and kinetics (Marcus theory): *Nanotech* 26, 354003 (2015); *Electrochim. Acta* 209, 498 (2016). I am not affiliated with this group.

Ignoring batteries, the next question is whether this work is of wide interest to the theoretical and/or electrochemical communities. The observation that each molecular configuration gives rise to a different instantaneous vertical IP must be generally true of all long-range electron transfer processes, including systems with simple redox species, not just for the rather ill-defined battery electrolyte oxidation problem. How to fold this predicted distribution into something that can compare with measurements is a fascinating and important question, one that must surely take into account electron transfer rates. I think that the authors have raised this interesting point, but have not done sufficient follow-through.

On the other hand, as far as impact on the computational community goes, I don't think the authors have advanced the state of the art sufficiently to justify publishing this work in Nature Communication, in my opinion.

My recommendation is to publish this work in a purely computational journal like JCTC. I would reconsider this recommendation if the authors can substantially revise their manuscript to address *general electrochemistry* at a molecular level. Assuming they continue their focus on vertical IP, I must express doubt whether this work will have much impact on the *battery* community in particular.

Reviewer #2 (Remarks to the Author):

The manuscript « Solvent-Anion Charge Transfer as Origin of Oxidative Degradation of Battery Electrolytes » of Fadel et al. deals with computational chemistry studies of various anion/solvent couples related to the field of Li-ion batteries. The authors show that the value of the ionization potential depends on the solvent/anion couple, and not only on one component. They also evidence

that the oxidative stability of the solvent/anion couple is often lower than that of each isolated species. Moreover, the authors propose a model to calculate the ionization potential of the solvent/anion system quite easily, by considering solvent/anion molecules in vacuum or in a solvent. This methodology can be applied to various solvent/anion systems of interest in the field of lithium-ion batteries. The obtained results also strongly suggest that hydrogen abstraction reactions from the solvent occurs after the formation of the charge-transfer complex formed upon oxidation.

It is a very detailed, careful and interesting work, proposing a new model, that is interesting for the Li-ion battery community and thus deserves to be published in Nature Communications.

I have some comments prior to publication:

1. Table 1: Ionization potentials are calculated depending on different functionals. Reference values for the various ionization potentials of interest should also be included in this table to allow for comparison (experimental values, and also values from very precise calculations). Can also some error bars be included in the displayed values? What is the accuracy on the given IP values? In the TFSI- +DME3 system, it is clear from the last 3 functionals used that the electron is removed from the solvent that has the highest IP value. The authors should also comment more at this stage on this point that is not necessarily intuitive.
2. Page 11, « IP study of anion-solvent pairs ». The gas phase IP values of TDI- and ACN have to be given in the text.
3. Page 12. Is there a trend between the geometry (the configuration) and the corresponding calculated IP value? Can this be rationalized? Can the authors be more precise than « 30 configurations »
4. Page 13. How can the delta value be rationalized? Where does this 2.8 eV value come from?
5. Table 2. The agreement between the values is generally good, except for PF₆-/PC, BF₄-/ACN and TFSI-/ACN. Can the authors comment on these particular cases?
6. Pages 23-24. The dielectric constants for the different solvents are not given in the text, but at 300 K, they are roughly equal to 5 for DME ; 47 for DMSO ; 66 for PC and 37 for ACN. So, delta/epsilon (with delta equal to 2.8) ranges between 0.04 and 0.07 eV for DMSO, PC and ACN, instead of the 0.1 to 0.15 eV values reported by the authors. I do not understand this discrepancy with a factor of 2.

Minor points to be changed:

End of page 11 : « CLARIFY »!!

Figure 3 is small and thus difficult to see

Reference 25 is not correct

Reviewer #3 (Remarks to the Author):

This computational study investigates electrochemical stability of liquid electrolytes comprising salt and solvent. Identifying electrolytes that are stable is extremely important in order to develop durable batteries with long cycle life. The electrochemical stability, as well as chemicals stability, of the electrolyte is an important parameter in this respect. While several recent theoretical studies have attempted to screen electrolytes by evaluating electrochemical stability, these studies have often used rather crude models that neglect the impact of solvents. This manuscript addresses this important gap and presents a detailed analysis of the oxidative stability of ions in presence of explicit solvent molecules. The results indicate that the oxidative stability of the electrolyte is governed by coupling

between anion and solvent and that the ionization potential depends on the specific chemistry of the components. Based on the calculations, the authors present a model to predict oxidative stability. Certainly, the manuscript adds scientific value in this highly populated research area. However, the study does not present experimental validation and various aspects of this computational study lack details. In particular, the authors should address the following comments:

1. Classical molecular dynamics was used to generate sufficient number of structures. Additional minimization of the MD-generated structures is not performed, and therefore it is extremely important to start with energetically favored configurations. The authors have cited that the details are found in a prior work where a completely different system was modeled with OPLS force field. Additional details regarding the validity of this model should be presented. In particular, OPLS is a non-polarizable force fields and therefore might not produce reliable anion-solvent structures. Furthermore, it is not clear if counterions were simulated and if so, how these were neglected in selecting the anion-solvent structures.
2. In order to gather reliable statistics, 30 different anion-solvent structures were used from the MD run. The authors should comment on why 30 structures are sufficient to generate reliable distribution of the ionization potential. For example, Figure 2a presents the distribution of IP for solvent-ion pairs, which is fairly broad. However, the distribution does not have a tail and it is not clear if calculations spanning a wider range of configurations would capture a wider distribution.
3. The study does not cite or present experimental data to validate the calculated IP.

Solvent-Anion Charge Transfer as Origin of Oxidative Degradation of Battery Electrolytes

Eric R. Fadel^{†,‡,||}, Francesco Faglioni[¶], Georgy Samsonidze[†], N. Molinari^{†,‡}, Boris V. Merinov[‡], William A. Goddard III[‡], Jeffrey C. Grossman^{||}, J.P. Mailoa[†], and B. Kozinsky^{†‡}

[†] Robert Bosch LLC, Research and Technology Center, Cambridge, Massachusetts 02139, USA,

[‡] John A. Paulson School of Engineering and Applied Sciences, Harvard University, Cambridge, MA 02138, USA.

^{||} Department of Materials Science and Engineering, Massachusetts Institute of Technology, Cambridge, MA 02139, USA

[¶] Department of Chemical and Geological Sciences, University of Modena and Reggio Emilia, Via Campi 103, 41125 Modena, Italy

[‡] Materials and Process Simulation Center, California Institute of Technology, Pasadena, CA 91125, USA

B.Kozinsky: bkoz@seas.harvard.edu, +1 617 252 0040

Response to the reviewers

We appreciate the time and effort that the editor and the reviewers have invested in reviewing our manuscript. We address the comments and questions raised by each reviewer below. **The green text** refers to what was present in the old version of the manuscript, while **the blue text** indicates changes or additions to the revised manuscript. All page numbers refer to the revised version of the manuscript.

Reviewer 1

We thank the reviewer for taking the time to read our manuscript and for providing detailed feedback. We address his/her concerns point by point below.

Reviewer: The authors apply DFT and other computational techniques to study issues related to oxidative stability of solvent molecules and salt (anions). They compute vertical ionization potentials (IP) and find that cooperative effects, i.e., charge-transfer between solvent and salt, may lower the IP.

My main concern is that this paper is mostly of interest to the computational community. If the paper was aimed at the battery community, I believe it has failed to stake claim to much relevance. I don't believe there is a single comparison with experimental values. The main problem is that the authors focus on vertical IP, which is generally not what is measured. The experimentally relevant quantity is clearly the oxidation potential, which is related to the

adiabatic (atom-relaxed) IP. Having read this paper, I am unable to come to a conclusion regarding whether any of the issues discussed are important for battery operations. (If the authors had listed the predicted oxidation potentials, the reader could have look at whether the issues raised are relevant by looking at whether the predicted oxidation potentials fall in the range relevant to cathode oxide materials.)

So, as a paper about batteries, I definitely recommend against publishing this work in Nature Communications. Incidentally, the authors should cite works by the Borodin group about the effects of different molecular configurations on oxidation, and kinetics (Marcus theory): *Nanotech* 26, 354003 (2015); *Electrochim. Acta* 209, 498 (2016). I am not affiliated with this group.

Author reply: We wish to address these concerns one by one. Firstly, it is not clear that adiabatic IP has any more relevance to experimental observations than vertical IP. Indeed, adiabatic IP calculations are atom-relaxed, which means that they do not take into account initial configuration sampling, and therefore completely neglect the effect of temperature. When authors do try to take into account initial configuration sampling as in the work by Borodin and coworkers¹, which was kindly brought to our attention by the reviewer, much fewer configurations are used. Furthermore these calculations still relax the final state, meaning that there is little reason to compare the adiabatic IP between these configurations. On the contrary, one of the aims of this work is to consider the effect of geometry on the oxidation, and in particular how temperature enables electron transfer at lower voltage.

In fact, this work focuses on the onset of the oxidation, for which the electron transfer is the important mechanism. On the other hand, when computing the adiabatic IP, one cannot separate the electron transfer from the subsequent relaxation and reaction mechanism. This issue leads to the incorrect interpretation of the weakening of the solvent in the presence of an anion as a consequence of oxidative degradation reaction¹⁻⁴, which we argue against in our work. Indeed, we show consistently that this effect is present without considerations of any subsequent relaxations or reactions, and stems from a much more fundamental behavior which is the possibility to form a charge transfer pair between the anion and the solvent.

We note also that we are not the first or the only ones to have focused on vertical IP to study oxidation stability. Others have successfully used a combination of molecular dynamics to sample configurations of electrolytes^{5,6} and different quantum chemistry methods to study oxidative stability of different chemistries. Other studies for ionic liquids^{7,8} use ab-initio MD and compute the difference between HOMO and LUMO energy levels, which is closely related to the vertical IP. In fact, these works find that the vertical IP is a good indicator for the oxidative stability of electrolytes. Miller and coworkers⁶ show that vertical IP distributions enable the study of solvent reorganization through linear response theory, which is an important aspect of oxidation mechanisms. Krylov and coworkers⁵ study vertical IP calculations of aqueous phenol and

phenolate and are able to show reasonable agreement with their experimental findings. It is important to mention here that they use valence photo-emission measurements using microjets and synchrotron radiation to precisely validate their computed vertical IP. Jónsson and Johansson⁹ perform a comparative oxidative stability of different anions with different DFT functionals, also using the vertical IP. Comparing against oxidation potentials found in literature, they find that vertical IP is a fair way to calculate stability, provided a good choice of (hybrid) functional is made.

For all of the above reasons, we maintain that the study based on vertical IP yields interesting properties that can be of interest to the whole battery community. In fact we show in this article a clear fundamental physical explanation of the phenomenon of solvent weakening that was already found in other computational methods as well as experimentally. Thus we are convinced that this effect is experimentally relevant, and argue that our computational method of choice (studying vertical IP distributions using a specifically chosen functional) is key to correctly understanding this effect.

The next important point to address is how theoretically computed properties relate to experimental measurements. First of all, oxidation potential measurements are rather difficult and imprecise, and there are large variations for the reported values from cyclic voltammetry measurements due to the effects of scan rates, different concentrations used, different electrodes, or even different methods to infer the onset oxidation voltage. Furthermore, there are many effects that are unaccounted for or poorly understood in most computational studies such as surface reactions that may have significant effects on the onset of the oxidation. Therefore, we believe that comparing computational results with experimentally obtained oxidation voltages is not a well defined criterion to judge computational studies. In particular, we do not believe that adiabatic IP is the best descriptor of the oxidation. Taking the oxidation voltage to be the energy it takes to remove an electron and place it at the Fermi level of the electrode, this quantity is more easily related to the instantaneous vertical IP for each given geometric configuration, as shown in the next graphic. Finally, we mention in passing that many computational papers find similar level of agreement with oxidation voltage measurements, whether these study adiabatic or vertical IP. Previous work on one of these anion-solvent pairs has shown vertical IP to be a good indicator of oxidation stability, and good agreement with oxidation voltages obtained from activation energies and experimental results¹⁰. Furthermore, in most cases, the difference between adiabatic and vertical IP is small (less than 1 eV¹), which is a spread smaller than oxidation voltage measurements. In general, there is no clear consensus in the community on the best theoretical method to describe oxidation, and the field requires more fundamental investigation and careful characterization.

We do however compare with trends found in literature, and we are particularly interested in showing how our interpretation of oxidation can explain both situations where the oxidation voltage depends on the choice of the anion and where it does not affect it. We compare with the work of Ue et al^{11,12} since it has many solvent-anion couples, but more importantly uses the same condition consistently across all chemistries. We find in particular the effect of saturation, where the oxidation potential saturates as the anion is changed to increasingly "stronger" anions. In our model, this is due to a switch between oxidizing the anion to oxidizing the solvent, and they do indeed find that PC oxidation plays a role in their observed oxidation potential.

Finally, we wish to show that for solvent gas phase IP which are available and unambiguous, computational vertical IP can match experimental values very well. We present new results in a table (added to the SI) showing that DLPNO-CCSD(T) IP match these experimental values very well, and we also note that the M06-HF IP values agree relatively well also for the chemistries considered in this work.

Original:

To provide supporting evidence for our hypothesis we note that the computed trend matches well the experimental observations by Ue et al¹² that for a given solvent (PC), the combined system oxidation potential depends on the anion IP, but that this dependence saturates for anions such as BF_4^- or stronger. Indeed, in their experimental results, the oxidation voltage for BF_4^- , PF_6^- and

AsF₆⁻ are roughly the same.

Changed to (pages 16-17):

For further supporting evidence to our hypothesis, we compare our results with previous experimental and computational works from the literature. Comparison between computed IP and experimental IP as measured in gas phase is provided in the supplementary information. Focusing on the trends of oxidation voltage with respect to the choice of anion-solvent chemistry, we argue that the observed behavior will depend on which species is oxidized. When considering electrolytes with the same solvent species and varying anion species, the oxidation voltage will increase with increasingly “strong” anion (while the species that is oxidized is the anion, i.e. it is “weak” compared to the solvent), until it saturates when switching happens and the solvent becomes the oxidized species. Therefore we expect a dependence of the oxidation voltage on the anion, but only for anions that are “weak” enough. When keeping the same anion and changing the solvent species, two different effects are at play to determine the oxidation voltage: the oxidation mechanism (whether the anion or the solvent is oxidized) and the solvation effect, which leads to a more ambiguous, chemistry-dependent trend. Looking at the experimental observations by Ue *et al.*^{11,12}, they first find that for a given solvent, (PC), the combined system oxidation potential depends on the anion IP, but that this dependence saturates for anions such as BF₄⁻ or stronger. Indeed, in their experimental results, the oxidation voltage for BF₄⁻, PF₆⁻ and AsF₆⁻ are roughly the same, which in our understanding of oxidation, hints at a switching from anion oxidation to solvent oxidation. Then, they use a more oxidation-resistant solvent, glutaronitrile, in order to determine the anodic stability order of those anions. Thus, we find a situation where both anion and solvent oxidation matter, depending on which species is oxidized, and their study shows the two possible behaviors of oxidation voltage with changing anion. All these observations are consistent and can be explained with our new understanding of oxidation. However, direct comparison to oxidation voltages as obtained through cyclic voltammetry measurements is not a well defined validation procedure for computational methods. These measurements have a large variance since they are affected by a wide range of parameters such as the scanning rate¹², the nature of the electrode, the concentration of species in the electrolyte, and even the method adopted to infer the oxidation voltage from the raw data¹³. In this work, emphasis was given to accurately describe oxidation mechanisms from the microscopic, ab-initio standpoint, compromising on the accuracy on the absolute value of the IP (also, surface effects are not taken into account). Whilst, for these reasons, quantitative comparison with experimental results is difficult, we believe that the trends presented in this work give important insights into the possible oxidation scenarios.

Original:

We examined this effect using more accurate DLPNO-CCSD(T) calculations of the (TDI⁻, PC), (TFSI⁻, PC), (BF₄⁻, PC) and (PF₆⁻, PC) combinations,

and found good agreement with the DFT results. For these calculations, the IP of the isolated species are 5.0 eV (TDI^-), 6.7 eV (TFSI^-), 8.6 eV (BF_4^-), 8.9 eV (PF_6^-) and 11 eV (PC). For the paired combinations we computed IP of 20 configurations each, from the same MD snapshots as used in M06-HF computations. The average IP values were found to be 4.9 eV for (TDI^- , PC), 6.6 eV for (TFSI^- , PC), 8.2 eV for (BF_4^- , PC) and 8.3 eV for (PF_6^- , PC). Similarly to the M06-HF DFT results and Equation 1, the anion is oxidized in the (TDI^- , PC) and (TFSI^- , PC) cases, with the pair IP close to the anion IP (since the difference between isolated anion and solvent IP is greater than δ). At the same time, the solvent is oxidized in the (BF_4^- , PC) and (PF_6^- , PC) cases with an average pair IP equal to the solvent IP minus δ . Thus, higher-order DLPNO-CCSD(T) computations show excellent agreement with the DFT conclusions, and validate our approach. The range of IP pair values due to geometry effects is similarly within about 1 eV. This indicates that configuration geometry effects are large compared to the energy difference between the M06-HF and CCSD(T) approximations.

Changed to (page 16):

We examined this effect using more accurate DLPNO-CCSD(T) calculations for the (TDI^- , PC), (TFSI^- , PC), (BF_4^- , PC) and (PF_6^- , PC) combinations, and the full results are reported in the supplementary information (tables S8 and S9). Briefly, we find not only that using a highly accurate level of theory yields IP values that closely agree with experimental data, but also that the trends presented in this work remain unaltered (see table S9). This validates our computational approach with regards to the choice of the DFT functional.

Added to the SI (pages 20-22):

DLPNO-CCSD(T) calculations were performed to validate our computational approach and our findings. These calculations were performed as described in the methods section of the main paper. Firstly, we show that vertical IP calculations using DLPNO-CCSD(T) calculations agree very well with experimental IP values for solvents. Furthermore, these results allow us to benchmark M06-HF for the chemistries studied. We find that our DFT calculations perform well for the chemistries studied. These results are summarized in table S8.

Table S8: DLPNO-CCSD(T) vertical IP for the 4 solvents and 4 anions, averaged over 25 configurations. Columns 1 and 5 show the different solvent and anion species respectively. Column 2 shows experimental values for solvents IP¹⁴. Columns 3 and 6 are M06-HF IP values for solvents and anions respectively. Columns 4 and 7 are DLPNO-CCSD(T) IP values for solvents and anions respectively.

Solvent	Exp. IP	M06-HF IP	CCSD(T) IP	Anion	M06-HF IP	CCSD(T) IP
DMSO	9.0-9.1	9.0	8.8	TDI ⁻	5.7	5.0
DME	9.8-9.9	10.6	9.9	TFSI ⁻	7.3	6.7
PC	-	12.0	11.0	BF ₄ ⁻	9.4	8.6
ACN	12.2-12.5	12.4	12.2	PF ₆ ⁻	10.2	8.9

Thus, it is possible to accurately depict the IP of isolated species using a more costly computational method. Furthermore, when using this very reliable computational method, the trends presented in this work are exactly the same. To prove this, we performed the same type of calculations as in the main paper, focusing on the anion-solvent pairs with PC solvent. We compute the average $IP_{\Delta SCF}$ over 20 configurations (using the same configurations as for the M06-HF analysis) and compare with our DFT calculations. The results are presented in table S9.

Table S9: DLPNO-CCSD(T) vertical $IP_{\Delta SCF}$ for the couples (PC,TDI⁻), (PC,TFSI⁻), (PC,BF₄⁻) and (PC,PF₆⁻) (rows 3, 4, 5 and 6). Columns 2, 3 and 4 show the isolated PC, isolated anion and anion-solvent pair IP from DLPNO-CCSD(T) calculations for the different couples. Columns 5, 6 and 7 show the same for M06-HF in comparison.

Method	CCSD(T) IP			M06-HF IP		
	PC	Anion	Pair	PC	Anion	Pair
PC+TDI ⁻	11	5.0	4.9	11.7	5.6	5.9
PC+TFSI ⁻	11	6.7	6.6	11.7	7.2	7.5
PC+BF ₄ ⁻	11	8.6	8.2	11.7	8.8	8.7
PC+PF ₆ ⁻	11	8.9	8.3	11.7	9.7	9.0

Similarly to the M06-HF DFT results, the anion is oxidized in the (TDI⁻, PC) and (TFSI⁻, PC) cases, with the pair IP close to the anion IP (since the difference between isolated anion and solvent IP is greater than δ). At the same time, the solvent is oxidized in the (BF₄⁻, PC) and (PF₆⁻, PC) cases with an average pair IP equal to the solvent IP minus $\delta = 2.8eV$. Thus, higher-order DLPNO-CCSD(T) computations show excellent agreement with the DFT conclusions, and validate our approach. The range of IP pair values due to geometry effects is similarly within about 1eV. This indicates that configuration geometry effects are large compared to the energy difference between the M06-HF and CCSD(T) approximations.

Reviewer: Ignoring batteries, the next question is whether this work is of wide interest to the theoretical and/or electrochemical communities. The observation that each molecular configuration gives rise to a different instantaneous vertical IP must be generally true of all long-range electron transfer processes, including systems with simple redox species, not just for the rather ill-defined battery electrolyte oxidation problem. How to fold this predicted distribution into something that can compare with measurements is a fascinating and important question, one that must surely take into account electron transfer rates. I think that the authors have raised this interesting point, but have not done sufficient follow-through.

On the other hand, as far as impact on the computational community goes, I don't think the authors have advanced the state of the art sufficiently to justify publishing this work in Nature Communication, in my opinion.

My recommendation is to publish this work in a purely computational journal like JCTC. I would reconsider this recommendation if the authors can substantially revise their manuscript to address *general electrochemistry* at a molecular level. Assuming they continue their focus on vertical IP, I must express doubt whether this work will have much impact on the *battery* community in particular.

Author reply: The goal of our work is to provide a new fundamental physical understanding of the coupling between the different species in the electrolyte with regard to its overall oxidation stability. As such, it is aimed to address some of the main questions arising in the Li-ion battery community, specifically addressing quantum-chemical mechanisms that limit battery stability and energy density. We believe our work is indeed well suited to address these questions in a way that would be of interest to all of the Li-ion battery community.

Reviewer 2

We thank the Reviewer for his/her enthusiastic review and support for our manuscript. Below we carefully answer the questions raised point-by-point.

Reviewer: The manuscript Solvent-Anion Charge Transfer as Origin of Oxidative Degradation of Battery Electrolytes of Fadel et al. deals with computational chemistry studies of various anion/solvent couples related to the field of Li-ion batteries. The authors show that the value of the ionization potential depends on the solvent/anion couple, and not only on one component. They also evidence that the oxidative stability of the solvent/anion couple is often lower than that of each isolated species. Moreover, the authors propose a model to calculate the ionization potential of the solvent/anion system quite easily, by considering solvent/anion molecules in vacuum or in a solvent. This methodology can be applied to various solvent/anion systems of interest in the field of lithium-ion batteries. The obtained results also strongly suggest that hydrogen abstrac-

tion reactions from the solvent occurs after the formation of the charge-transfer complex formed upon oxidation.

It is a very detailed, careful and interesting work, proposing a new model, that is interesting for the Li-ion battery community and thus deserves to be published in Nature Communications.

I have some comments prior to publication: 1. Table 1: Ionization potentials are calculated depending on different functionals. Reference values for the various ionization potentials of interest should also be included in this table to allow for comparison (experimental values, and also values from very precise calculations). Can also some error bars be included in the displayed values? What is the accuracy on the given IP values? In the TFSI+DME3 system, it is clear from the last 3 functionals used that the electron is removed from the solvent that has the highest IP value. The authors should also comment more at this stage on this point that is not necessarily intuitive.

Author reply: We add a reference to the table in the supplementary information with all our DLPNO-CCSD(T) calculated IP for the isolated species, with experimental values when available (we opt to keep all these values in that table for clearer presentation). We also add these values in the caption for the table. DFT methods do not supply error bars, which is an inherent problem of all iterative minimization methods. Therefore we cannot include error bars for those calculated IP. Once convergence of the method is achieved, the accuracy of DFT calculations is therefore better defined as the error on the calculated value compared to the experimental value or the best computational method available. This comparison is done in more detail in the supplementary information where we show that DLPNO-CCSD(T) IP values agree very well with experimental values when available, and that comparing these values from very accurate computational methods to our chosen functional M06-HF, we find that the M06-HF IP are relatively precise for the chemistries considered in this work. Finally, as suggested, we add a comment on the counter-intuitive fact that the species that is oxidized is the solvent with higher IP, to improve the discussion.

Added in the manuscript (page 10):

Finally, we note that in the case of TFSI⁻ and three DME, the functionals that completely oxidize one molecule oxidize the solvent, which has a higher IP than the anion. This is counter-intuitive, and we discuss in detail in the rest of this work how charge-transfer pair formation is the cause of this oxidation mechanism.

Original:

We examined this effect using more accurate DLPNO-CCSD(T) calculations of the (TDI⁻, PC), (TFSI⁻, PC), (BF₄⁻, PC) and (PF₆⁻, PC) combinations, and found good agreement with the DFT results. For these calculations, the IP of the isolated species are 5.0 eV (TDI⁻), 6.7 eV (TFSI⁻), 8.6 eV (BF₄⁻), 8.9 eV (PF₆⁻) and 11 eV (PC). For the paired combinations we computed IP of 20 configurations each, from the same MD snapshots as used in M06-HF

computations. The average IP values were found to be 4.9 eV for (TDI⁻, PC), 6.6 eV for (TFSI⁻, PC), 8.2 eV for (BF₄⁻, PC) and 8.3 eV for (PF₆⁻, PC). Similarly to the M06-HF DFT results and Equation 1, the anion is oxidized in the (TDI⁻, PC) and (TFSI⁻, PC) cases, with the pair IP close to the anion IP (since the difference between isolated anion and solvent IP is greater than δ). At the same time, the solvent is oxidized in the (BF₄⁻, PC) and (PF₆⁻, PC) cases with an average pair IP equal to the solvent IP minus δ . Thus, higher-order DLPNO-CCSD(T) computations show excellent agreement with the DFT conclusions, and validate our approach. The range of IP pair values due to geometry effects is similarly within about 1 eV. This indicates that configuration geometry effects are large compared to the energy difference between the M06-HF and CCSD(T) approximations.

Changed to (page 16):

We examined this effect using more accurate DLPNO-CCSD(T) calculations for the (TDI⁻, PC), (TFSI⁻, PC), (BF₄⁻, PC) and (PF₆⁻, PC) combinations, and the full results are reported in the supplementary information (tables S8 and S9). Briefly, we find not only that using a highly accurate level of theory yields IP values that closely agree with experimental data, but also that the trends presented in this work remain unaltered (see table S9). This validates our computational approach with regards to the choice of the DFT functional.

Added to the SI (pages 20-22):

DLPNO-CCSD(T) calculations were performed to validate our computational approach and our findings. These calculations were performed as described in the methods section of the main paper. Firstly, we show that vertical IP calculations using DLPNO-CCSD(T) calculations agree very well with experimental IP values for solvents. Furthermore, these results allow us to benchmark M06-HF for the chemistries studied. We find that our DFT calculations perform well for the chemistries studied. These results are summarized in table S8.

Table S8: DLPNO-CCSD(T) vertical IP for the 4 solvents and 4 anions, averaged over 25 configurations. Columns 1 and 5 show the different solvent and anion species respectively. Column 2 shows experimental values for solvents IP¹⁴. Columns 3 and 6 are M06-HF IP values for solvents and anions respectively. Columns 4 and 7 are DLPNO-CCSD(T) IP values for solvents and anions respectively.

Solvent	Exp. IP	M06-HF IP	CCSD(T) IP	Anion	M06-HF IP	CCSD(T) IP
DMSO	9.0-9.1	9.0	8.8	TDI ⁻	5.7	5.0
DME	9.8-9.9	10.6	9.9	TFSI ⁻	7.3	6.7
PC	-	12.0	11.0	BF ₄ ⁻	9.4	8.6
ACN	12.2-12.5	12.4	12.2	PF ₆ ⁻	10.2	8.9

Thus, it is possible to accurately depict the IP of isolated species using a more costly computational method. Furthermore, when using this very reliable computational method, the trends presented in this work are exactly the same.

To prove this, we performed the same type of calculations as in the main paper, focusing on the anion-solvent pairs with PC solvent. We compute the average $IP_{\Delta SCF}$ over 20 configurations (using the same configurations as for the M06-HF analysis) and compare with our DFT calculations. The results are presented in table S9.

Table S9: DLPNO-CCSD(T) vertical $IP_{\Delta SCF}$ for the couples (PC, TDI⁻), (PC, TFSI⁻), (PC, BF₄⁻) and (PC, PF₆⁻) (rows 3, 4, 5 and 6). Columns 2, 3 and 4 show the isolated PC, isolated anion and anion-solvent pair IP from DLPNO-CCSD(T) calculations for the different couples. Columns 5, 6 and 7 show the same for M06-HF in comparison.

Method	CCSD(T) IP			M06-HF IP		
Couples	PC	Anion	Pair	PC	Anion	Pair
PC+TDI ⁻	11	5.0	4.9	11.7	5.6	5.9
PC+TFSI ⁻	11	6.7	6.6	11.7	7.2	7.5
PC+BF ₄ ⁻	11	8.6	8.2	11.7	8.8	8.7
PC+PF ₆ ⁻	11	8.9	8.3	11.7	9.7	9.0

Similarly to the M06-HF DFT results, the anion is oxidized in the (TDI⁻, PC) and (TFSI⁻, PC) cases, with the pair IP close to the anion IP (since the difference between isolated anion and solvent IP is greater than δ). At the same time, the solvent is oxidized in the (BF₄⁻, PC) and (PF₆⁻, PC) cases with an average pair IP equal to the solvent IP minus $\delta = 2.8eV$. Thus, higher-order DLPNO-CCSD(T) computations show excellent agreement with the DFT conclusions, and validate our approach. The range of IP pair values due to geometry effects is similarly within about 1eV. This indicates that configuration geometry effects are large compared to the energy difference between the M06-HF and CCSD(T) approximations.

Reviewer: 2. Page 11, IP study of anion-solvent pairs. The gas phase IP values of TDI- and ACN have to be given in the text.

3. Page 12. Is there a trend between the geometry (the configuration) and the corresponding calculated IP value? Can this be rationalized? Can the authors be more precise than 30 configurations

Author reply: For the first point, we have updated the figure to include more detailed histograms from 200 configurations for the two cases of anion oxidation ((TFSI⁻, PC) pair) and solvent ((PF₆⁻, DME) pair) oxidation. We also added in the text the gas phase IP for TFSI⁻, PF₆⁻, PC and DME to add clarity to the discussion, as suggested.

The second point is a very interesting question which led us to perform an additional investigation and significantly expand the manuscript. In particular, the question of identifying geometry internal variables that correlate well with the IP of a configuration is an important question but most likely requires

many more calculations. However, from our calculations, we already have good hints as to what separates low-IP configurations and high-IP configurations on energy grounds. Firstly, from our analysis on the energy distributions of the MD snapshots, we know that the energy distribution of these configurations are of gaussian-type (therefore there is no bias in sampling the IP distribution). We include a discussion on this matter in the supplemental material, where we do the following. Taking advantage of our 200 configuration calculations for the (TFSI⁻, PC) pair (for which the anion is oxidized) and the (PF₆⁻, DME) pair (for which the solvent is oxidized), we plot two heat maps for each pair: the energy of the initial and oxidized with respect to the IP. These energies are relative to the average neutral (respectively oxidized) energy. We find that the two cases are very different in nature. For the case where the anion is oxidized, we find that for configurations with IP around the IP peak, both initial and final energy are near their average. Low-IP configurations have high initial energy and low final energies, while high-IP configurations have low to average initial energy but high final energies. Since in this case the anion is oxidized, we can hypothesize that these correlations are due to the small variations of anion geometries, and that initial and final energy are both distributed as gaussians, with the IP distributions being consequently distributed as their difference. For the case where the solvent is oxidized, we find a very different behavior, where there is little correlation between the IP and initial energy, but a large correlation with final energy. Since the final energy is the energy of the charge transfer complex, we can find a very simple reason to this correlation: the IP of a configuration and its final energy are both dependent on the dipole energy, unlike the initial state. Therefore, in the case of solvent oxidation, IP configuration is fully determined by the geometry through the dipole energy of the charge transfer complex.

Original:

In this section, we present the results for the IP study of anion-solvent pairs, and provide a simple empirical formula for the IP of the pair, before proposing a simple physical model in the next section. First, we find that the spread of the IP values over all the snapshot configurations is significant (on the order of 1 eV), across all chemistries. Figure 1.a shows a histogram of IP of 30 different configurations for a specific pair, (TDI⁻, ACN), with the dotted lines representing the average IP of the isolated species TDI⁻ which is a "weak" anion (i.e., has a low IP in gas phase) and ACN which is a "strong" solvent (i.e., has a high IP).

Figure 2: Example of the detailed study for one pair: (TDI^- , ACN). a). Histogram of the pair IP over the 30 configurations, with the blue dashed line representing the isolated anion IP and the red dashed line the isolated solvent IP. b). Predicted IP from our electrostatics model against the computed $\text{IP}_{\Delta\text{SCF}}$ for all 30 configurations. Supplementary material reports a full study for all anion-solvent pairs (figures S3 and S4).

Changed to (pages 11-13):

In this section, we present the results for the IP study of anion-solvent pairs, and provide a simple empirical formula for the IP of the pair, before proposing a simple physical model in the next section. First, we find that the spread of the IP values over all the snapshot configurations is significant (on the order of 1 eV), across all chemistries. To investigate this spread we study two specific couples, (TFSI^- , PC) and (PF_6^- , DME), using 200 configurations of the anion-solvent pair. Figure 2 shows the obtained distributions of vertical IP. The first pair comprises TFSI^- that is a “weak” anion (i.e., has an IP in gas phase of 7.23 eV), and PC that is a “strong” solvent (i.e., has an IP of 12.66

eV). The second pair comprises a “strong” anion (PF_6^- with an average IP of 9.7 eV), and a “weak” solvent (DME with an average IP of 10.2 eV). We indeed find that the spread is significant, but that the average distribution converges for a moderate number of configurations (for $(\text{TFSI}^-, \text{PC})$ for example, the average with 30 random configurations is 7.5 eV and the average with 200 random configurations is 7.45 eV). The use of 30 random configurations is deemed a good compromise between computational cost and accuracy, and therefore applied to the study of all anion-solvent pairs.

We also plot the heat maps of the initial and final energies with respect to the ionization potential (Figure 2) for these two anion-solvent pairs. We find that there is a distinction between anion-solvent pairs for which the anion is oxidized, and those for which the solvent is oxidized. For the $(\text{TFSI}^-, \text{PC})$ pair, the anion is the species that is oxidized. In this case, for configurations with IP near the average the IP peak, the configuration’s initial and final energies are near the average of their respective distributions. For configurations corresponding to the lower end of the IP distribution, their initial energy is high relative to the set of all configurations and their final energy is low. Inversely, configurations with high IP have low to average initial energy but high final energies. Considering the $(\text{PF}_6^-, \text{DME})$ pair in which the solvent is oxidized, we find a very different behavior: the initial energy is relatively uncorrelated to the IP of the configuration, and the distribution of IP is mostly governed by the final energy. This difference is significant and highlights a difference in the oxidation mechanism which can be understood using the model discussed later in this work.

Figure 2: IP distribution for 200 random configurations of anion-solvent pair. The first column corresponds to the (TFSI⁻, PC) pair, and the second column corresponds to the (PF₆⁻, DME) pair. Supplementary information reports a full study for all anion-solvent pairs (figure S3) with 30 configurations. The color code, which is consistent in all this work, represents in blue configurations where the anion is fully oxidized (as is the case for all configurations of the (TFSI⁻, PC) pair) and in red configurations where the solvent is oxidized (as is the case for all configurations of the (PF₆⁻, DME) pair). Below these two figures, we plot the heat maps of the initial (middle plot) and final (bottom plot) energies for the two pairs. The energies are plotted relative to the average energy (either initial or final).

Reviewer: 4. Page 13. How can the delta value be rationalized? Where does this 2.8 eV value come from?

Author reply: As discussed in the article, δ is due to dipole interaction. The interesting question asked here is why does 2.8 eV work as an average over these different chemistries. Excluding TDI⁻ for which there is no solvent weak enough to have solvent oxidation, we have three anions for which solvent oxidation happens for some choice of the solvents studied. From our interpretation of δ as dipole energy, we should conclude that this value must be the average dipole formation energy over the phase space of configurations, which means that if for two electrolytic systems, the average anion-solvent distance and charges are roughly the same, the value of δ would be similar. This seems to be true for our systems of small molecule anions and solvents. It is clear however that there are no reasons to a priori use $\delta = 2.8\text{eV}$ for any electrolyte chemistry. In particular, even in cases where the average dipole energy between one anion and the closest

solvent was the same, the change in solvent will affect the value of δ_{\bullet} in solution, as discussed in the subsection on solvation effects.

Reviewer: 5. Table 2. The agreement between the values is generally good, except for PF₆⁻/PC, BF₄⁻/ACN and TFSI⁻/ACN. Can the authors comment on these particular cases?

Author reply: This is indeed an interesting question, and the reasons are not entirely clear yet. The absence of quantum coupling contributions to the energy of the anion-solvent complex is the only assumption entering the model. We find that for these cases with small anions, this hypothesis is good for the majority of cases but breaks down to some extent. It is a somewhat idealized model which we use to rationalize the charge transfer effect. We rephrase this in the main text for clarity.

Original:

The most significant deviation of the model from the Δ SCF result occurs for configurations involving PC and strong anions, (BF₄⁻, PC) and (PF₆⁻, PC). This is possibly due to the fact that these anions are compact and may approach the solvent closely enough for the quantum-classical energy decoupling to be incomplete, since this is the only approximation in the model.

Changed to (page 20):

The most significant deviation of the model from the Δ SCF result occurs for configurations involving PC and strong anions, (BF₄⁻, PC) and (PF₆⁻, PC). The only approximation in the model is the absence of coupling in quantum energy between the anion and solvent, therefore these deviations are possibly due to anion-solvent coupling, given that these anions are compact and may approach the solvent closely enough.

Reviewer: 6. Pages 23-24. The dielectric constants for the different solvents are not given in the text, but at 300 K, they are roughly equal to 5 for DME ; 47 for DMSO ; 66 for PC and 37 for ACN. So, δ/ϵ (with δ equal to 2.8) ranges between 0.04 and 0.07 eV for DMSO, PC and ACN, instead of the 0.1 to 0.15 eV values reported by the authors. I do not understand this discrepancy with a factor of 2.

Author reply: We rephrase this discussion to make it clearer. We rewrite the discussion in the main text and make a table in the supplementary information to summarize our calculations results. We use implicit solvation of the anion-solvent pairs to estimate the value of δ_{\bullet} , and compare to our approximation $\frac{\delta}{\epsilon}$. The values that we mention are between 0.1 to 0.15 eV are actually the computed values of δ_{\bullet} , and as pointed out by the reviewer, there is some discrepancy for the higher dielectric constant solvents (although the differences are on the order of 0.05-0.07 eV). As mentioned in the discussion, the probable explanation for these discrepancies is that we used fewer configurations to estimate δ_{\bullet} (5 configurations), or the accuracy of our input value for δ (this is an average for all pairs studied in this work).

Original:

To understand the trend, we introduce the effect of solvation into the model of Equation 3, treating it as an effect of classical dielectric continuum on the energy of point charges. The computational finding that δ decreases in solvated systems is expected, because it represents the classical electrostatic energy of the dipole formation, which should decrease with increasing dielectric constant (denoted by ϵ). The detailed derivation of the effect of full solvation in the value of the dipole stabilization energy δ is given in the supplemental information. Following the discussion the supplementary information (see equation S4), the value of the electrostatic stabilization energy at full solvation is given by $\delta_{\bullet} = \delta/\epsilon$. Thus the value of δ_{\bullet} in PCM-solvation for DME is much closer to the vacuum value of than for the other three solvents, which we expect given that DME has a much lower dielectric constant. In fact we find that the value of δ_{\bullet} can be approximated even without PCM calculations and only using the equation above starting with the value of δ computed in vacuum and the dielectric constant ϵ of the solvent. When we compare the values of δ_{\bullet} from PCM calculations to δ/ϵ (where the vacuum value δ is about 2.8 eV) averaged over 5 configurations chosen close to the IP distribution peak, we find that the difference ranges from 0.03 to 0.07 eV. The conclusion is that we have a practical rapid recipe for estimating IP of the full solvent-anion system that requires only computations of IP of individual solvated species.

Changed to (pages 25-26):

To analyze the long-range effect of full solvation, and in particular the change in δ_{\bullet} , we employ the PCM implicit solvent model for all the BF_4^- pairs (i.e., BF_4^- solvated with DMSO, DME, PC or ACN) with dielectric constants of 46.8 for DMSO, 4.2 for DME, 65.5 for PC, and 35.7 for ACN. We find that the solvent is still the oxidized species, just like in the vacuum case, and the value of the difference δ_{\bullet} between the IP of the solvent and that of the solvent-anion pair is indeed lower but still significant in the PCM-solvated calculations. The values of δ_{\bullet} are reported in table S7 of the supplementary information. To understand the trend, we introduce the effect of solvation into the model of Equation 3, treating it as an effect of classical dielectric continuum on the energy of point charges. The computational finding that δ decreases in solvated systems is expected, because it represents the classical electrostatic energy of the dipole formation, which should decrease with increasing dielectric constant (denoted by ϵ). The detailed derivation of the effect of full solvation in the value of the dipole stabilization energy δ is given in the supplemental information. Following the discussion the supplementary information (see equation S4), the value of the electrostatic stabilization energy at full solvation is given by $\delta_{\bullet} = \delta/\epsilon$. Thus the value of δ_{\bullet} in PCM-solvation for DME is much closer to the vacuum value of than for the other three solvents, which we expect given that DME has a much lower dielectric constant. In fact we find that the value of δ_{\bullet} can be approximated even without PCM calculations and only using the equation above starting with the value of δ computed in vacuum and the

dielectric constant ϵ of the solvent. Table S7 shows the comparison between this approximation and the computed value for δ_{\bullet} . When we compare the values of δ_{\bullet} from PCM calculations to δ/ϵ (where the vacuum value δ is about 2.8 eV) averaged over 5 configurations chosen close to the IP distribution peak, we find that the difference ranges from 0.03 to 0.07 eV. Therefore we have a practical, fast recipe for estimating IP of the full solvent-anion system that requires only computations of IP of individual solvated species.

Added to SI (pages 19-20):

For all couples with the BF_4^- anion, we compute the value of δ_{\bullet} with 5 configurations of anion-solvent pairs and using implicit solvation to approximate the full solvation of this pair. The dielectric constants for the solvents are taken to be 46.8 for DMSO, for 4.2 DME, 65.5 for PC and 35.7 for ACN. We find that the solvent is still the oxidized species, just like in the vacuum case, and the value of the difference δ_{\bullet} between the IP of the solvent and that of the solvent-anion pair is indeed lower but still significant in the PCM-solvated calculations. The results are presented in the table S7. We find that we can relatively well approximate δ_{\bullet} by $\frac{\delta}{\epsilon}$, taking the value of δ to be 2.8 eV. The discrepancies are larger for the solvents with lower dielectric constants, (but still on the order of 0.05-0.07 eV). This could be because we have not sampled enough configurations to compute δ_{\bullet} , or because instead of using the specific value of δ for every chemistry, we used the 2.8 eV value which is averaged over all the pairs we studied in this work (although the differences are very small, they are also on the order of the discrepancies we observe here).

Table S7: Values of δ_{\bullet} for all pairs with BF_4^- anion. δ_{\bullet} is defined as the IP drop that is the difference of the pair IP (in implicit solvation) and the IP of the lone solvent (in implicit solvation) in the case where the solvent is oxidized (column 1). This is averaged over 5 different configurations picked close to the IP distribution peak for the pair (and in the case of ACN, only configurations leading to solvent oxidation). Column 2 represents the value of $\frac{\delta}{\epsilon}$ for the same couples, taking δ to be 2.8 eV and ϵ to be the dielectric constants for the solvents as described above.

Couples	δ_{\bullet}	$\frac{\delta}{\epsilon}$
BF_4^- +DMSO	0.09	0.05
BF_4^- +DME	0.61	0.63
BF_4^- +PC	0.13	0.05
BF_4^- +ACN	0.16	0.09

Reviewer: Minor points to be changed: End of page 11 : CLARIFY !! Figure 3 is small and thus difficult to see Reference 25 is not correct

Author reply: We thank the reviewer for spotting these errors, which we corrected. Figure 3 was replotted larger to make it more readable.

Reviewer 3

We thank the Referee for his/her overall positive review and support for our manuscript. Below we answer the questions raised point-by-point.

Reviewer: This computational study investigates electrochemical stability of liquid electrolytes comprising salt and solvent. Identifying electrolytes that are stable is extremely important in order to develop durable batteries with long cycle life. The electrochemical stability, as well as chemicals stability, of the electrolyte is an important parameter in this respect. While several recent theoretical studies have attempted to screen electrolytes by evaluating electrochemical stability, these studies have often used rather crude models that neglect the impact of solvents. This manuscript addresses this important gap and presents a detailed analysis of the oxidative stability of ions in presence of explicit solvent molecules. The results indicate that the oxidative stability of the electrolyte is governed by coupling between anion and solvent and that the ionization potential depends on the specific chemistry of the components. Based on the calculations, the authors present a model to predict oxidative stability. Certainly, the manuscript adds scientific value in this highly populated research area. However, the study does not present experimental validation and various aspects of this computational study lack details. In particular, the authors should address the following comments:

1. Classical molecular dynamics was used to generate sufficient number of structures. Additional minimization of the MD-generated structures is not performed, and therefore it is extremely important to start with energetically favored configurations. The authors have cited that the details are found in a prior work where a completely different system was modeled with OPLS force field. Additional details regarding the validity of this model should be presented. In particular, OPLS is a non-polarizable force fields and therefore might not produce reliable anion-solvent structures. Furthermore, it is not clear if counterions were simulated and if so, how these were neglected in selecting the anion-solvent structures.

Author reply: This question is indeed important as one of the premises of this study is the sampling of configurations from MD to study more reliably the stability of the electrolyte. To answer this question, we conducted a systematic verification of the force-field: we use 200 configurations for the (TFSI, PC) pair, and compare the MD and DFT (M06-HF) energies. The plot is presented in the updated figure 1. We find that the two gaussian-type distributions are close, with the DFT energy being more spread out. There are no bias of the MD, and by fitting the gaussian curves and looking at the ration of the variances of the two curves (that we link to the square of the temperature ratio), we find an effective temperature of the DFT energy distribution of 380 K compared to the 300 K temperature of the MD distribution. Thus, we conclude that the MD samples configurations well, and in particular does not induce a biased sampling of the IP distribution.

Secondly, all the MD simulations contained a large number of solvent molecules, and one anion-cation couple, for charge neutrality. We found the cation always solvated far away from the anion, therefore not affecting the extraction of the anion-solvent complexes. The choice of focusing on the anion-solvent complex was made in consideration the fact that anion-cation-solvent clusters are more stable than anion-solvent clusters (for example³), and therefore the presence of the cation is not critical for the study of the oxidative stability of the system.

Original:

The structures are obtained from snapshots of classical molecular dynamics simulations. First, the anion of interest and the solvent molecules are placed on the vertices of a three-dimensional cubic grid, with the aim to create a low-density non-overlapping initial structure. Once generated, the structures are brought close to equilibrium by a series of energy minimization, compression/decompression, and annealing stages, broadly based on previous works¹⁵, to overcome local energy barriers in search of lower energy minima, and, ultimately, more representative structures. The structures are then evolved using the velocity-Verlet algorithm, and with a constant time step of 1.0 fs, in the NPT ensemble. Temperature and pressure are kept at 300 K and 1 atmosphere, respectively, with the use of a Nosé-Hoover barostat and thermostat¹⁶⁻¹⁸. The coordinates of all atoms are saved every 10^3 timesteps (i.e. 1 picosecond) to ensure sufficiently uncorrelated structures. In post processing the positions of the anion and of the closest X solvent molecules (X ranging from one to five) are extracted from the structure snapshots and used in the ab-initio calculations. The structures do not undergo any further geometry optimization, instead we sample different configurations of this system in order to gather statistics on the IP of anion and solvent complexes. All molecular dynamics simulations are performed in the LAMMPS simulation package¹⁹. The interactions are modeled using the OPLS2005 force-field²⁰ from Schrodinger Inc. Figure 1 shows an example of a snapshot for the solvated anion (for clarity, not all solvent molecules are shown), in the case of the (TFSI⁻, DME) pair, as well as examples of extracted configurations with one anion and its closest solvent for the same pair.

Figure 1: Examples of configurations taken for the (TFSI⁻, DME) pair. The top picture is the solvated anion. The bottom pictures are examples of structures used to perform the DFT study.

Changed to (pages 7-8):

Previous works have shown the importance of sampling electrolyte configurations^{1,5-7}, and have done so using classical MD^{5,6} or ab-initio MD⁷. In this work, the structures are obtained from snapshots of classical MD simulations, obtained as follows.

First, the anion of interest and the solvent molecules are placed on the vertices of a three-dimensional cubic grid, with the aim to create a low-density non-overlapping initial structure. Once generated, the structures are brought close to equilibrium by a series of energy minimization, compression/decompression, and annealing stages, broadly based on previous works^{15,21}, to overcome local energy barriers in search of lower energy minima, and, ultimately, more repre-

sentative structures. The structures are then evolved using the velocity-Verlet algorithm, with a time step of 1.0 fs, in the constant number of atoms, pressure, and temperature (or NPT) ensemble. Temperature and pressure are kept at 300 K and 1 atmosphere, respectively, with a Nosé-Hoover barostat and thermostat¹⁶⁻¹⁸. The coordinates of all atoms are saved every 10^3 timesteps (i.e., 1 ps) to ensure sufficiently uncorrelated structures. In post processing the positions of the anion and of the closest X solvent molecules (X ranging from one to five) are extracted from the snapshots and used in the ab-initio calculations. The structures do not undergo any further geometry optimization, instead we sample different configurations of this system in order to gather statistics on the IP of anion and solvent complexes. All molecular dynamics simulations are performed in the LAMMPS simulation package¹⁹. The interactions are modeled using the OPLS2005 force-field²⁰ from Schrodinger Inc. Since the MD structures do not undergo further geometry optimization, it is important to ensure that the configurations are well sampled by the classical energy model approximation. To this end, we consider a set of 200 configurations of the (TFSI⁻, PC) pair and plot in Figure 1(a) the energy for these configurations computed with the classical force-field (red) and with DFT (blue). The two distributions are Gaussian-like, with the classical force-field underestimating the energy of the configurations compared to DFT, but preserving the distribution of the energy in the phase space. Thus we conclude that the OPLS2005 force-field samples configurations reasonably. Figure 1(b) shows an example of a snapshot for the solvated anion (for clarity, not all solvent molecules are shown), in the case of the (TFSI⁻, DME) pair, and Figure 1(c) to (e) examples of extracted configurations with one anion and its closest solvent for the same pair.

Figure 1: Figure (a) represents for 200 configurations of anion-solvent pairs from MD snapshots for the (TFSI⁻, PC) pair, the distribution in energy computed from MD and the distribution of energy computed from DFT (M06-HF). Both these distributions are plotted relative to their average. (b) shows a solvated anion for an MD snapshot in the case of the (TFSI⁻, DME) pair. (c) to (e) show examples of configurations of solvent-anion pair from MD snapshots for the (TFSI⁻, DME) pair.

Reviewer: 2. In order to gather reliable statistics, 30 different anion-solvent structures were used from the MD run. The authors should comment on why 30 structures are sufficient to generate reliable distribution of the ionization potential. For example, Figure 2a presents the distribution of IP for solvent-ion pairs, which is fairly broad. However, the distribution does not have a tail and it is not clear if calculations spanning a wider range of configurations would capture a wider distribution.

Author reply: This is also an important question. For our study, we used the averages of the IP distributions to analyze and discuss the trends we found. However, we do believe that all of the distribution is important if one wants to completely understand the oxidation of the electrolyte system, and especially the low-IP configurations. By performing IP calculations for 200 configurations for two different pairs ((TFSI⁻, PC) and (PF₆⁻, DME)), we obtain more accurately sampled IP distributions in the two oxidation scenarii (anion or solvent oxidized). We find that the average is converged for 30 configurations, but indeed, we find slightly wider distributions with more configurations. Should one pur-

sue an accurate study and be interested in the low-IP tail of the distribution, we can conclude that many configurations should be sampled to do this analysis (we also point to our answer to one of reviewer 2's question where we discuss interesting correlation of configuration IP with initial and final energy).

Original:

In this section, we present the results for the IP study of anion-solvent pairs, and provide a simple empirical formula for the IP of the pair, before proposing a simple physical model in the next section. First, we find that the spread of the IP values over all the snapshot configurations is significant (on the order of 1 eV), across all chemistries. Figure 1.a shows a histogram of IP of 30 different configurations for a specific pair, (TDI^- , ACN), with the dotted lines representing the average IP of the isolated species TDI^- which is a "weak" anion (i.e., has a low IP in gas phase) and ACN which is a "strong" solvent (i.e., has a high IP).

Figure 2: Example of the detailed study for one pair: (TDI^- , ACN). a). Histogram of the pair IP over the 30 configurations, with the blue dashed line representing the isolated anion IP and the red dashed line the isolated solvent IP. b). Predicted IP from our electrostatics model against the computed $\text{IP}_{\Delta\text{SCF}}$ for all 30 configurations. Supplementary material reports a full study for all anion-solvent pairs (figures S3 and S4).

Changed to (pages 11-13):

In this section, we present the results for the IP study of anion-solvent pairs, and provide a simple empirical formula for the IP of the pair, before proposing a simple physical model in the next section. First, we find that the spread of the IP values over all the snapshot configurations is significant (on the order of 1 eV), across all chemistries. To investigate this spread we study two specific couples, (TFSI^- , PC) and (PF_6^- , DME), using 200 configurations of the anion-solvent pair. Figure 2 shows the obtained distributions of vertical IP. The first pair comprises TFSI^- that is a “weak” anion (i.e., has an IP in gas phase of 7.23 eV), and PC that is a “strong” solvent (i.e., has an IP of 12.66

eV). The second pair comprises a “strong” anion (PF_6^- with an average IP of 9.7 eV), and a “weak” solvent (DME with an average IP of 10.2 eV). We indeed find that the spread is significant, but that the average distribution converges for a moderate number of configurations (for $(\text{TFSI}^-, \text{PC})$ for example, the average with 30 random configurations is 7.5 eV and the average with 200 random configurations is 7.45 eV). The use of 30 random configurations is deemed a good compromise between computational cost and accuracy, and therefore applied to the study of all anion-solvent pairs.

We also plot the heat maps of the initial and final energies with respect to the ionization potential (Figure 2) for these two anion-solvent pairs. We find that there is a distinction between anion-solvent pairs for which the anion is oxidized, and those for which the solvent is oxidized. For the $(\text{TFSI}^-, \text{PC})$ pair, the anion is the species that is oxidized. In this case, for configurations with IP near the average the IP peak, the configuration’s initial and final energies are near the average of their respective distributions. For configurations corresponding to the lower end of the IP distribution, their initial energy is high relative to the set of all configurations and their final energy is low. Inversely, configurations with high IP have low to average initial energy but high final energies. Considering the $(\text{PF}_6^-, \text{DME})$ pair in which the solvent is oxidized, we find a very different behavior: the initial energy is relatively uncorrelated to the IP of the configuration, and the distribution of IP is mostly governed by the final energy. This difference is significant and highlights a difference in the oxidation mechanism which can be understood using the model discussed later in this work.

Figure 2: IP distribution for 200 random configurations of anion-solvent pair. The first column corresponds to the (TFSI⁻, PC) pair, and the second column corresponds to the (PF₆⁻, DME) pair. Supplementary information reports a full study for all anion-solvent pairs (figure S3) with 30 configurations. The color code, which is consistent in all this work, represents in blue configurations where the anion is fully oxidized (as is the case for all configurations of the (TFSI⁻, PC) pair) and in red configurations where the solvent is oxidized (as is the case for all configurations of the (PF₆⁻, DME) pair). Below these two figures, we plot the heat maps of the initial (middle plot) and final (bottom plot) energies for the two pairs. The energies are plotted relative to the average energy (either initial or final).

Reviewer: 3. The study does not cite or present experimental data to validate the calculated IP.

Author reply: The primary goal of the work is to develop fundamental understanding of the oxidation mechanism of complex electrolytes, using an approximate DFT-based approach and making sure to validate the mechanisms with higher-order quantum chemistry methods. The choice of the M06-HF functional in particular was made to correctly describe charge localization upon oxidation, but comes at the cost of less precise IP values. However to answer the question of validating our method, we use one of the most accurate quantum chemistry methods, DLPNO-CCSD(T), and relate this to experimental values. We have added a clearer description of this work in the supplementary information and refer to it in the main paper. We show that DLPNO-CCSD(T) IP values match very well experimental IP values when these are available. Furthermore, comparing with the M06-HF IP values, we find that these agree relatively well with the DLPNO-CCSD(T) values for the chemistries we looked at. Secondly, for the

four anion-solvent pairs with the PC solvent, we repeated our study using 20 configurations and the DLPNO-CCSD(T) method. We find that the conclusions of our work still hold, even when the IP values between the two quantum chemistry methods are different. The IP_{fit} expression still holds, and even the value of δ is the same. The species that is oxidized is also consistent between the two methods. Therefore, we are able to show that using a very accurate quantum chemistry method, we can find IP values that agree very well with experiments, and furthermore, using this method to perform the same study of anion-solvent pair IP gives the same conclusions.

Original:

We examined this effect using more accurate DLPNO-CCSD(T) calculations of the (TDI⁻, PC), (TFSI⁻, PC), (BF₄⁻, PC) and (PF₆⁻, PC) combinations, and found good agreement with the DFT results. For these calculations, the IP of the isolated species are 5.0 eV (TDI⁻), 6.7 eV (TFSI⁻), 8.6 eV (BF₄⁻), 8.9 eV (PF₆⁻) and 11 eV (PC). For the paired combinations we computed IP of 20 configurations each, from the same MD snapshots as used in M06-HF computations. The average IP values were found to be 4.9 eV for (TDI⁻, PC), 6.6 eV for (TFSI⁻, PC), 8.2 eV for (BF₄⁻, PC) and 8.3 eV for (PF₆⁻, PC). Similarly to the M06-HF DFT results and Equation 1, the anion is oxidized in the (TDI⁻, PC) and (TFSI⁻, PC) cases, with the pair IP close to the anion IP (since the difference between isolated anion and solvent IP is greater than δ). At the same time, the solvent is oxidized in the (BF₄⁻, PC) and (PF₆⁻, PC) cases with an average pair IP equal to the solvent IP minus δ . Thus, higher-order DLPNO-CCSD(T) computations show excellent agreement with the DFT conclusions, and validate our approach. The range of IP pair values due to geometry effects is similarly within about 1 eV. This indicates that configuration geometry effects are large compared to the energy difference between the M06-HF and CCSD(T) approximations.

Changed to (page 16):

We examined this effect using more accurate DLPNO-CCSD(T) calculations for the (TDI⁻, PC), (TFSI⁻, PC), (BF₄⁻, PC) and (PF₆⁻, PC) combinations, and the full results are reported in the supplementary information (tables S8 and S9). Briefly, we find not only that using a highly accurate level of theory yields IP values that closely agree with experimental data, but also that the trends presented in this work remain unaltered (see table S9). This validates our computational approach with regards to the choice of the DFT functional.

Added in SI (pages 20-22):

DLPNO-CCSD(T) calculations were performed to validate our computational approach and our findings. These calculations were performed as described in the methods section of the main paper. Firstly, we show that vertical IP calculations using DLPNO-CCSD(T) calculations agree very well with experimental IP values for solvents. Furthermore, these results allow us to benchmark M06-HF for the chemistries studied. We find that our DFT calculations perform well for the chemistries studied. These results are summarized in table S8.

Table S8: DLPNO-CCSD(T) vertical IP for the 4 solvents and 4 anions, averaged over 25 configurations. Columns 1 and 5 show the different solvent and anion species respectively. Column 2 shows experimental values for solvents IP¹⁴. Columns 3 and 6 are M06-HF IP values for solvents and anions respectively. Columns 4 and 7 are DLPNO-CCSD(T) IP values for solvents and anions respectively.

Solvent	Exp. IP	M06-HF IP	CCSD(T) IP	Anion	M06-HF IP	CCSD(T) IP
DMSO	9.0-9.1	9.0	8.8	TDI ⁻	5.7	5.0
DME	9.8-9.9	10.6	9.9	TFSI ⁻	7.3	6.7
PC	-	12.0	11.0	BF ₄ ⁻	9.4	8.6
ACN	12.2-12.5	12.4	12.2	PF ₆ ⁻	10.2	8.9

Thus, it is possible to accurately depict the IP of isolated species using a more costly computational method. Furthermore, when using this very reliable computational method, the trends presented in this work are exactly the same. To prove this, we performed the same type of calculations as in the main paper, focusing on the anion-solvent pairs with PC solvent. We compute the average $IP_{\Delta SCF}$ over 20 configurations (using the same configurations as for the M06-HF analysis) and compare with our DFT calculations. The results are presented in table S9.

Table S9: DLPNO-CCSD(T) vertical $IP_{\Delta SCF}$ for the couples (PC,TDI⁻), (PC,TFSI⁻), (PC,BF₄⁻) and (PC,PF₆⁻) (rows 3, 4, 5 and 6). Columns 2, 3 and 4 show the isolated PC, isolated anion and anion-solvent pair IP from DLPNO-CCSD(T) calculations for the different couples. Columns 5, 6 and 7 show the same for M06-HF in comparison.

Method	CCSD(T) IP			M06-HF IP		
	PC	Anion	Pair	PC	Anion	Pair
PC+TDI ⁻	11	5.0	4.9	11.7	5.6	5.9
PC+TFSI ⁻	11	6.7	6.6	11.7	7.2	7.5
PC+BF ₄ ⁻	11	8.6	8.2	11.7	8.8	8.7
PC+PF ₆ ⁻	11	8.9	8.3	11.7	9.7	9.0

Similarly to the M06-HF DFT results, the anion is oxidized in the (TDI⁻, PC) and (TFSI⁻, PC) cases, with the pair IP close to the anion IP (since the difference between isolated anion and solvent IP is greater than δ). At the same time, the solvent is oxidized in the (BF₄⁻, PC) and (PF₆⁻, PC) cases with an average pair IP equal to the solvent IP minus $\delta = 2.8eV$. Thus, higher-order DLPNO-CCSD(T) computations show excellent agreement with the DFT conclusions, and validate our approach. The range of IP pair values due to geometry effects is similarly within about 1eV. This indicates that configuration geometry effects are large compared to the energy difference between the M06-HF and CCSD(T) approximations.

References

- [1] Borodin, O.; Olguin, M.; Spear, C. E.; Leiter, K. W.; Knap, J. *Nanotechnology* **2015**, *26*, 354003.
- [2] Borodin, O.; Jow, T. R. *ECS Transactions* **2011**, *33*, 77–84.
- [3] Borodin, O.; Behl, W.; Jow, T. R. *The Journal of Physical Chemistry C* **2013**, *117*, 8661–8682.
- [4] Kim, D. Y.; Park, M. S.; Lim, Y.; Kang, Y.-S.; Park, J.-H.; Doo, S.-G. *Journal of Power Sources* **2015**, *288*, 393–400.
- [5] Ghosh, D.; Roy, A.; Seidel, R.; Winter, B.; Bradforth, S.; Krylov, A. I. *The Journal of Physical Chemistry B* **2012**, *116*, 7269–7280.
- [6] Barnes, T. A.; Kaminski, J. W.; Borodin, O.; Miller III, T. F. *The Journal of Physical Chemistry C* **2015**, *119*, 3865–3880.
- [7] Ong, S. P.; Andreussi, O.; Wu, Y.; Marzari, N.; Ceder, G. *Chemistry of Materials* **2011**, *23*, 2979–2986.
- [8] Zhang, Y.; Shi, C.; Brennecke, J. F.; Maginn, E. J. *The Journal of Physical Chemistry B* **2014**, *118*, 6250–6255.
- [9] Jónsson, E.; Johansson, P. *Physical Chemistry Chemical Physics* **2015**, *17*, 3697–3703.
- [10] Faglioni, F.; Merinov, B. V.; Goddard, W. A.; Kozinsky, B. *Physical Chemistry Chemical Physics* **2018**, *20*, 26098–26104.
- [11] Ue, M.; Takeda, M.; Takehara, M.; Mori, S. *Journal of The Electrochemical Society* **1997**, *144*, 2684–2688.
- [12] Ue, M.; Murakami, A.; Nakamura, S. *Journal of The Electrochemical Society* **2002**, *149*, A1572–A1577.
- [13] Xu, K. *Chemical reviews* **2004**, *104*, 4303–4418.
- [14] Linstrom, P. J.; Mallard, W. G. *Journal of Chemical & Engineering Data* **2001**, *46*, 1059–1063.
- [15] Molinari, N.; Khawaja, M.; Sutton, A.; Mostofi, A. *The Journal of Physical Chemistry B* **2016**, *120*, 12700–12707.
- [16] Hoover, W. G. *Physical review A* **1985**, *31*, 1695.
- [17] Nosé, S. *The Journal of chemical physics* **1984**, *81*, 511–519.
- [18] Hoover, W. G. *Physical Review A* **1986**, *34*, 2499.

- [19] Plimpton, S.; Crozier, P.; Thompson, A. *Sandia National Laboratories* **2007**, *18*.
- [20] Banks, J. L. et al. *Journal of Computational Chemistry* **2005**, *26*, 1752–1780.
- [21] Molinari, N.; Mailoa, J. P.; Kozinsky, B. *Chemistry of Materials* **2018**, *30*, 6298–6306.

Reviewers' comments:

Reviewer #1 (Remarks to the Author):

In the resubmitted manuscript, the authors did not comparison their predictions with experimental redox potentials. Granted the latter can be problematic to measure but other battery modeling papers have included such comparisons. Scientific research does not take place in a vacuum; there are community standards. Without such comparisons, the authors' conclusions,

"we find that hydrogen abstraction ... is not the primary reason for the electrolyte's weakening ... but is more likely to subsequent occur after a charge-transfer complex is formed upon oxidation ..."

however reasonable, is basically completely unsupported. In fact, the authors never even explored computationally whether the charge-transfer complex will equilibrate into something different (e.g., a non-charge-transfer complex) from their non-adiabatic configuration.

As such, the paper will have little impact in the battery field. I am happy to review this as a purely computational/electrochemical paper below.

=====
=====

The strength of the paper is the use of sophisticated DFT and quantum chemistry techniques to cross-check electron (hole) localization. The use of molecular dynamics (MD) trajectory configurations is also useful, but these should have been applied to the product state too. This and other weaknesses should be addressed.

I will marginally support publishing this work in Nature Communications if the following changes are made (at a minimum), *and* if other reviewers are enthusiastic about this work. This is clearly at the editor's discretion.

1. Change the title to something more appropriate. The paper does not adequately address the origin of battery electrolyte oxidation. The title should say this is a battery-inspired, high-level electronic structure work.

2. Cite the proper literature in this field, e.g., from the Sprik and Warshel groups (I am not affiliated):
J. Phys. Chem. Lett. 3, 3411 (2012)
J. Phys. Chem. B 112, 257 (2008)
J. Am. Chem. Soc. (1987)

3. In at least one case where the authors find unusual charge-transfer complexes in vertical ionization energy (VIE) calculations, perform a similar calculation in the product state (with configurations chosen from MD trajectories after solvent molecules are allowed to thermally equilibrate to the oxidized complex).

4. Acknowledge that the authors' VIE, and the HOMO-LUMO language in general, is an approximation and that redox should be modeled in more rigorous ways. Cite Energy Envir. Sci. 11, 2306 by Peljo and Girault, "Electrochemical potential window of battery electrolytes: the HOMO-LUMO misconception". (That title speaks for itself.)

5. A minor point: on page 16 and SI table S8: comparison of predictions and experiments are in SI mentioned. What experimental IP are those in Ref. 20? Photoemission? Please specify.

More details on the above, which also touch on the authors' rebuttal letter.

1. This point does not need further elaboration.
2. The authors calculate non-adiabatic ionization potential (or vertical ionization energy, VIE) throughout. As their Ref. 27 acknowledged, VIE is for comparing with photoemission. In the literature and battery community, electrochemically measured redox potentials are computed using adiabatic oxidation potentials.

The authors wrote in their letter that

"... [adiabatic] IP calculations are atom-relaxed, which means that they do not take into account initial configuration sampling, and therefore completely neglect the effect of temperature."

I find this statement shocking. Seminal DFTMD-based calculations of redox potentials, which take proper account of sampling and temperature, have been repeatedly reported by the Sprik group (and others). Classical MD calculations of redox potentials have an even longer history -- much of it credited to the group of Ariel Warshel, a recipient of the Nobel Prize for Chemistry. The authors try to use static configurations so they can apply more sophisticated DFT and/or quantum chemistry methods. This is an approximation of better sampling techniques already used in the literature. The authors must convey this message to the reader.

The Marcus Theory for electron transfer is also explicitly formulated at finite temperature. This is a well-known, Nobel-Prize winning work.

3. There seems no reason the authors cannot redo what they do in the initial (pre-charge-transfer) electronic surface for the product (post-transfer) electronic surface. This will help determine whether charge-transfer complexes seen in VIE conditions survive solvent reorganization following charge transfer.
4. The battery field has historically and erroneously used the HOMO-LUMO language not only for electrolytes, but also for cathode materials, even though internal reorganization (free) energies for organic carbonate molecules used in commercial batteries are large, and cathode materials are polaronic (not metallic) conductors. After the publication of Peljo and Girault (2018), even leading experimentalists in the field have acknowledged this language is an error (e.g., Dr. Kang Xu of Army Research Lab in his recent lectures).

To omit this significant advance/development in scientific understanding in this field, and to insist on using the old HOMO-LUMO language, does the reader a disservice.

Reviewer #2 (Remarks to the Author):

The authors have carefully answered to the questions raised by the reviewers and the paper now deserves to be published.

Reviewer #3 (Remarks to the Author):

The authors have appropriately responded to this reviewer's comments.

Role of Solvent-Anion Charge Transfer in Oxidative Degradation of Battery Electrolytes

Eric R. Fadel^{†,‡,||}, Francesco Faglioni[¶], Georgy Samsonidze[†], N. Molinari^{†,‡}, Boris V. Merinov[‡], William A. Goddard III[‡], Jeffrey C. Grossman^{||}, J.P. Mailoa[†], and B. Kozinsky^{†‡}

[†] Robert Bosch LLC, Research and Technology Center, Cambridge, Massachusetts 02139, USA,

[‡] John A. Paulson School of Engineering and Applied Sciences, Harvard University, Cambridge, MA 02138, USA.

^{||} Department of Materials Science and Engineering, Massachusetts Institute of Technology, Cambridge, MA 02139, USA

[¶] Department of Chemical and Geological Sciences, University of Modena and Reggio Emilia, Via Campi 103, 41125 Modena, Italy

[‡] Materials and Process Simulation Center, California Institute of Technology, Pasadena, CA 91125, USA

B.Kozinsky: bkoz@seas.harvard.edu, +1 617 252 0040

Response to the reviewer 1's concerns

We appreciate the time and effort that the editor and the reviewers have invested in reviewing our reply and changes. We address the concerns raised by reviewer 1 below. The blue text indicates changes or additions to the revised manuscript. All page numbers refer to the revised version of the manuscript. In the manuscript, we highlight all the changes (including rephrasing some sentences and minor corrections) in red.

Reviewer: In the resubmitted manuscript, the authors did not comparison their predictions with experimental redox potentials. Granted the latter can be problematic to measure but other battery modeling papers have included such comparisons. Scientific research does not take place in a vacuum; there are community standards. Without such comparisons, the authors' conclusions,

"we find that hydrogen abstraction ... is not the primary reason for the electrolyte's weakening ... but is more likely to subsequent occur after a charge-transfer complex is formed upon oxidation ..."

however reasonable, is basically completely unsupported. In fact, the authors never even explored computationally whether the charge-transfer complex will equilibrate into something different (e.g., a non-charge-transfer complex) from their non-adiabatic configuration.

As such, the paper will have little impact in the battery field. I am happy to review this as a purely computational/electrochemical paper below.

=====
The strength of the paper is the use of sophisticated DFT and quantum chemistry techniques to cross-check electron (hole) localization. The use of molecular dynamics (MD) trajectory configurations is also useful, but these should have been applied to the product state too. This and other weaknesses should be addressed.

I will marginally support publishing this work in Nature Communications if the following changes are made (at a minimum), *and* if other reviewers are enthusiastic about this work. This is clearly at the editor's discretion.

1. Change the title to something more appropriate. The paper does not adequately address the origin of battery electrolyte oxidation. The title should say this is a battery-inspired, high-level electronic structure work.

Author Reply: The title is changed to "Role of Solvent-Anion Charge Transfer in Oxidative Degradation of Battery Electrolytes".

Reviewer: 2. Cite the proper literature in this field, e.g., from the Sprik and Warshel groups (I am not affiliated): J. Phys. Chem. Lett. 3, 3411 (2012) J. Phys. Chem. B 112, 257 (2008) J. Am. Chem. Soc. (1987)

Reviewer: 4. Acknowledge that the authors' VIE, and the HOMO-LUMO language in general, is an approximation and that redox should be modeled in more rigorous ways. Cite Energy Envir. Sci. 11, 2306 by Peljo and Girault, "Electrochemical potential window of battery electrolytes: the HOMO-LUMO misconception". (That title speaks for itself.)

Author Reply: In order to clarify these approximations to the reader as suggested by the reviewer, we have added corrections to the discussion of the method on page 4 (lines 16 to 25):

When determining the stability window of an electrolyte, one should consider the rate of the oxidation (and reduction) reactions, which is linked to the activation free energy of the electron transfer. Computing this quantity accurately requires systematic explicit calculations of the reaction¹⁻³, and we have for example done this for the Li-PEO-TFSI system⁴. In this work, we focus on the onset of oxidation to estimate an approximate upper limit of the voltage stability window, therefore we do not look at possible degradation pathways and molecular geometry relaxation following the electron removal. Thus, instead of computing adiabatic ionization potential (IP), which approximates the redox potential, our approach of estimating the activation energy of the electron transfer relies on computing vertical IP distributions of molecular complexes to describe the weakest configurations, i.e., most susceptible to oxidation processes near the charged cathode. Vertical IP has been shown to be a useful

indicator of oxidative stability⁴⁻⁶. It is important for electron transfer to consider the full oxidation reaction instead of computing HOMO-LUMO levels of individual species⁷, which motivates our choice of computing vertical IP of explicit solvent-anion complexes, using the Δ SCF approach, as an energy difference between oxidized and initial states (with no geometry relaxation)^{5,8,9}. As mentioned before however, many different processes such as complex multi-step reactions with different molecules in the electrolyte, as well as surface effects, impact the true voltage window of stability⁷, which are not accounted for in vertical (or adiabatic) IP calculations, as the knowledge of degradation pathways and mechanisms is very limited. We note here that in our investigations we do not report results of complexes involving Li^+ cations, since these always have higher IP, as previously reported¹⁰, confirmed by our calculations and expected from electrostatic considerations. Thus, the most relevant configurations for oxidation, i.e., with the lowest IP, are those where the cation is not present. To study the redox behavior of these anion-solvent complexes, it is necessary to carefully choose the computational approach that most faithfully treats the ionization of whole systems of multiple molecules. Hence, we tested the most popular computational methods on vertical IPs.

Reviewer: 3. In at least one case where the authors find unusual charge-transfer complexes in vertical ionization energy (VIE) calculations, perform a similar calculation in the product state (with configurations chosen from MD trajectories after solvent molecules are allowed to thermally equilibrate to the oxidized complex).

Author Reply: In this work, we focused on the onset of the oxidation reaction, and since our aim is to study the maximum operating voltage of a battery, we emphasize the initial electron transfer and not the full complexity of the subsequent oxidation reaction. However, while we did not look at the subsequent decomposition reaction in full solvation, we did show how the anion-solvent system could possibly relax after the electron transfer in the case of charge transfer. As such, we already discuss in the main text that it seems hydrogen moves to reduce the dipole moment, which suggests that the charge transfer complex may not survive, even without considering solvent reorganization. In this paper we argue that the existence or lack of charge transfer complex will affect the subsequent oxidation reaction, but we expect that in general the charge transfer complex would not survive in solution (as indeed the reviewer is suggesting). We wish to stress again that this work is focused on the onset of the oxidation, that is the initial electron transfer, and that the charge transfer complex is important not as a final product of the oxidation (this is not true in general) but because it gives us insight into the weakening of the solvent-anion couple, which is the main finding in this paper. A more thorough study of the decomposition reaction mechanism for one of the electrolytes is performed in ref⁴.

Reviewer: 5. A minor point: on page 16 and SI table S8: comparison of predictions and experiments are in SI mentioned. What experimental IP are those in Ref. 20? Photoemission? Please specify.

Author Reply: Indeed these are photoemission spectroscopy measurements. We have added this information to the supplementary material.

Reviewer: More details on the above, which also touch on the authors' rebuttal letter.

1. This point does not need further elaboration.
2. The authors calculate non-adiabatic ionization potential (or vertical ionization energy, VIE) throughout. As their Ref. 27 acknowledged, VIE is for comparing with photoemission. In the literature and battery community, electrochemically measured redox potentials are computed using adiabatic oxidation potentials.

The authors wrote in their letter that

"... [adiabatic] IP calculations are atom-relaxed, which means that they do not take into account initial configuration sampling, and therefore completely neglect the effect of temperature."

I find this statement shocking. Seminal DFTMD-based calculations of redox potentials, which take proper account of sampling and temperature, have been repeatedly reported by the Sprik group (and others). Classical MD calculations of redox potentials have an even longer history – much of it credited to the group of Ariel Warshel, a recipient of the Nobel Prize for Chemistry. The authors try to use static configurations so they can apply more sophisticated DFT and/or quantum chemistry methods. This is an approximation of better sampling techniques already used in the literature. The authors must convey this message to the reader.

The Marcus Theory for electron transfer is also explicitly formulated at finite temperature. This is a well-known, Nobel-Prize winning work.

Author Reply: In the manuscript correction given above we aim to further clarify the intent and applicability of using vertical IP. Specifically in response to the comment, we wish to dispel what seems to be a misunderstanding by the reviewer. The sentence from our earlier reply that is quoted refers to the fact that adiabatic IP in itself holds no information on temperature. However, we certainly agree with the reviewer that temperature effects are taken into account in the Marcus theory of electron transfer, which has been applied using classical or ab-initio MD calculations of the initial and final state energy distributions, e.g. in the references suggested¹⁻³. We agree with the referee that our sampling method is an approximation, but we are focused on finding an upper limit of the voltage stability window, and not so much on predicting oxidation reaction rates, thus vertical IP is a good approximation to obtain trends. In the context of electrolyte stability, vertical or adiabatic IP can be used, and we have highlighted that many similar studies use one or the other, sometimes both^{5,6,11,12}.

Reviewer: 3. There seems no reason the authors cannot redo what they do in the initial (pre-charge-transfer) electronic surface for the product (post-transfer) elec-

tronic surface. This will help determine whether charge-transfer complexes seen in VIE conditions survive solvent reorganization following charge transfer.

Author Reply: This was answered above.

Reviewer: 4. The battery field has historically and erroneously used the HOMO-LUMO language not only for electrolytes, but also for cathode materials, even though internal reorganization (free) energies for organic carbonate molecules used in commercial batteries are large, and cathode materials are polaronic (not metallic) conductors. After the publication of Peljo and Girault (2018), even leading experimentalists in the field have acknowledged this language is an error (e.g., Dr. Kang Xu of Army Research Lab in his recent lectures).

To omit this significant advance/development in scientific understanding in this field, and to insist on using the old HOMO-LUMO language, does the reader a disservice.

Author Reply: Our corrections to the manuscript aim to address the referee’s concern. In response to the comment, we do not use HOMO-LUMO language to discuss our results but only to report published results based on HOMO-LUMO considerations. In fact, we express in the opening of the Methods section that we use DeltaSCF. To prevent misunderstandings, we added a sentence to explicitly remind that this consists in taking the difference between energies from two SCF computations, one for the reduced and one for the oxidized state. We agree that IPs, whether vertical or adiabatic, do not take into account the full mechanisms of the oxidation reaction and can only be used to provide approximations of the oxidation potentials.

References

- [1] Adriaanse, C.; Cheng, J.; Chau, V.; Sulpizi, M.; VandeVondele, J.; Sprik, M. *The journal of physical chemistry letters* **2012**, *3*, 3411–3415.
- [2] Ayala, R.; Sprik, M. *The Journal of Physical Chemistry B* **2008**, *112*, 257–269.
- [3] Hwang, J. K.; Warshel, A. *Journal of the American Chemical Society* **1987**, *109*, 715–720.
- [4] Faglioni, F.; Merinov, B. V.; Goddard, W. A.; Kozinsky, B. *Physical Chemistry Chemical Physics* **2018**, *20*, 26098–26104.
- [5] Ghosh, D.; Roy, A.; Seidel, R.; Winter, B.; Bradforth, S.; Krylov, A. I. *The Journal of Physical Chemistry B* **2012**, *116*, 7269–7280.
- [6] Barnes, T. A.; Kaminski, J. W.; Borodin, O.; Miller III, T. F. *The Journal of Physical Chemistry C* **2015**, *119*, 3865–3880.
- [7] Peljo, P.; Girault, H. H. *Energy & Environmental Science* **2018**, *11*, 2306–2309.

- [8] Marenich, A. V.; Ho, J.; Coote, M. L.; Cramer, C. J.; Truhlar, D. G. *Physical Chemistry Chemical Physics* **2014**, *16*, 15068–15106.
- [9] Husch, T.; Yilmazer, N. D.; Balducci, A.; Korth, M. *Physical Chemistry Chemical Physics* **2015**, *17*, 3394–3401.
- [10] Borodin, O.; Behl, W.; Jow, T. R. *The Journal of Physical Chemistry C* **2013**, *117*, 8661–8682.
- [11] Jónsson, E.; Johansson, P. *Physical Chemistry Chemical Physics* **2015**, *17*, 3697–3703.
- [12] Borodin, O.; Olguin, M.; Spear, C. E.; Leiter, K. W.; Knap, J. *Nanotechnology* **2015**, *26*, 354003.

Reviewers' comments:

Reviewer #1 (Remarks to the Author):

The authors have addressed most of my concerns except point 3. They appear to have misunderstood my concern.

The solvent relaxation issue I raised does not pertain to subsequent chemical reaction. It pertains to solvent relaxation immediately upon electron removal and possible charge transfer. See attached figure adapted from wikipedia. The blue arrow is what the authors calculate (see however below). The red arrow is the adiabatic oxidation potential. The green arrow is the solvent relaxation that can occur following vertical ionization, *within tens of picoseconds* (and does not have to involve chemical reactions). If during solvent relaxation (green) the charge transferred predicted by the authors goes away, their calculations are only relevant for those tens of picoseconds, too fast to measure or have impact on subsequent oxidation.

This is what the authors should state in the paper to clarify to the reader, or address with further calculations.

Incidentally, if a static dielectric continuum is used throughout, the calculation is not truly "vertical IP" because the outer shell solvent molecules should not have had time to relax (see above too); only the high frequency dielectric constant should have been used. This is another "approximation" that should be discussed in the manuscript.

If these are addressed I don't need to review the paper again.

[Redacted]

Role of Solvent-Anion Charge Transfer in Oxidative Degradation of Battery Electrolytes

Eric R. Fadel^{†,‡,||}, Francesco Faglioni[¶], Georgy Samsonidze[†], N. Molinari^{†,‡}, Boris V. Merinov[‡], William A. Goddard III[‡], Jeffrey C. Grossman^{||}, J.P. Mailoa[†], and B. Kozinsky^{†‡}

[†] Robert Bosch LLC, Research and Technology Center, Cambridge, Massachusetts 02139, USA,

[‡] John A. Paulson School of Engineering and Applied Sciences, Harvard University, Cambridge, MA 02138, USA.

^{||} Department of Materials Science and Engineering, Massachusetts Institute of Technology, Cambridge, MA 02139, USA

[¶] Department of Chemical and Geological Sciences, University of Modena and Reggio Emilia, Via Campi 103, 41125 Modena, Italy

[‡] Materials and Process Simulation Center, California Institute of Technology, Pasadena, CA 91125, USA

B.Kozinsky: bkoz@seas.harvard.edu, +1 617 252 0040

Response to the reviewer 1's concerns

We appreciate the time and effort that the editor and the reviewers have invested in reviewing our second reply and changes. We address the last concerns raised by reviewer 1 below. The blue text indicates changes or additions to the revised manuscript. All page numbers refer to the revised version of the manuscript. In the manuscript, we highlight all the changes (including rephrasing some sentences and minor corrections) in red.

Reviewer: The authors have addressed most of my concerns except point 3. They appear to have misunderstood my concern.

The solvent relaxation issue I raised does not pertain to subsequent chemical reaction. It pertains to solvent relaxation immediately upon electron removal and possible charge transfer. See attached figure adapted from wikipedia. The blue arrow is what the authors calculate (see however below). The red arrow is the adiabatic oxidation potential. The green arrow is the solvent relaxation that can occur following vertical ionization, *within tens of picoseconds* (and does not have to involve chemical reactions). If during solvent relaxation (green) the charge transferred predicted by the authors goes away, their calculations are only relevant for those tens of picoseconds, too fast to measure or have impact on subsequent oxidation.

This is what the authors should state in the paper to clarify to the reader, or address with further calculations.

Author Reply: We had indeed misunderstood the reviewer’s concern here. Although we argue that solvent relaxation is part of the many phenomena that lead to the decomposition of the electrolyte, it is true that after the electron removal, there is no way to ascertain how much impact would a dipole formation have on the subsequent processes. The H abstraction study presented in this paper is done in vacuum, and is only meant to illustrate one of possible scenario of how the charge-transfer complex could impact the subsequent reaction. We change the relevant part of the manuscript to make this clear to the reader following the reviewer’s suggestion.

Original:

Previous studies examined the possibility of hydrogen transfer after oxidation¹⁻³, suggesting that it is the reason for the weakening of the combined solvent-anion system. We have shown that the weakening effect of the anion-solvent pair can be explained regardless of any specific degradation steps following the system oxidation. However, the study of electrostatic intermolecular interactions shown above can give new insight into the oxidation-driven reaction mechanisms. Without doing an exhaustive study of reaction mechanisms and their energy barriers, this section focuses on the impact of the proton (ionic charge) transfer mechanism. We postulate that H transfer is energetically favorable in the cases of charge-transfer complexes partly because of electrostatics, since it would compensate the dipole formation and lower the electrostatic energy. In this work, using the same configuration of anion-solvent pairs, no spontaneous intermolecular reaction was observed when we relaxed the geometries. We proceeded to study H transfer by initially displacing H towards the anion, followed by relaxation of the oxidized structure. We found that in those anion-solvent pairs where the charge-transfer dipole was formed (i.e. solvent oxidized), an H atom from the solvent was observed to transfer to the anion in about 80% of the configurations. In all cases, if the structure was not oxidized, the H atom relaxed to the initial structure. For the anion-solvent pairs where oxidation results in electronic charge transfer, hydrogen transfer indeed lowers the dipole moment and total energy of the system. In the case of BF_4^- and PF_6^- anions, the hydrogen atom transfers to a fluorine, forming HF, leaving a BF_3 or PF_5 . Figure 5 shows typical snapshots of configurations with charge transfer. We conclude that hydrogen transfer is not the cause of the electrolyte weakening but rather a consequence of the intrinsic electronic charge-transfer complex formation, governed by the interplay between (quantum) ionization and (classical) electrostatic dipole energetics. A detailed research of degradation mechanisms and energy barriers in light of the findings of this work was performed for (TFSI, DME)⁴. Other combinations will be addressed in a future article.

Changed to (pages 22-24):

Previous studies examined the possibility of hydrogen transfer after oxidation¹⁻³, suggesting that it is the reason for the weakening of the combined solvent-anion system. We have shown that the weakening effect of the anion-

solvent pair can be explained regardless of any specific degradation steps following the system oxidation. However, the study of electrostatic intermolecular interactions shown above can give new insight into the oxidation-driven reaction mechanisms. Without doing an exhaustive study of reaction mechanisms and their energy barriers, this section focuses on the impact of the proton (ionic charge) transfer mechanism. Here we note that this study is done in vacuum, whereas the true degradation mechanism involves the coupling of different processes including solvent reorganization and molecular relaxation after the electron removal. Thus, this study only provides an example of possible evolution following the charge-transfer complex formation. We postulate that H transfer is energetically favorable in the cases of charge-transfer complexes partly because of electrostatics, since it would compensate the dipole formation and lower the electrostatic energy. In this work, using the same configuration of anion-solvent pairs, no spontaneous intermolecular reaction was observed when we relaxed the geometries. We proceeded to study H transfer by initially displacing H towards the anion, followed by relaxation of the oxidized structure. We found that in those anion-solvent pairs where the charge-transfer dipole was formed (i.e. solvent oxidized), an H atom from the solvent was observed to transfer to the anion in about 80% of the configurations. In all cases, if the structure was not oxidized, the H atom relaxed to the initial structure. For the anion-solvent pairs where oxidation results in electronic charge transfer, hydrogen transfer indeed lowers the dipole moment and total energy of the system. In the case of BF_4^- and PF_6^- anions, the hydrogen atom transfers to a fluorine, forming HF, leaving a BF_3 or PF_5 . Figure 5 shows typical snapshots of configurations with charge transfer. We conclude that hydrogen transfer is not the cause of the electrolyte weakening but rather a consequence of the intrinsic electronic charge-transfer complex formation, governed by the interplay between (quantum) ionization and (classical) electrostatic dipole energetics. A detailed research of degradation mechanisms and energy barriers in light of the findings of this work was performed for (TFSI, DME)⁴. Other combinations will be addressed in a future article.

Original:

This study shows that oxidative stability of Li-ion battery electrolytes is governed by non trivial coupling between anion and solvent and requires their coupling to be simulated explicitly. We find that only one molecule, either the solvent or the anion, loses an electron upon oxidation, but the value of the ionization potential (IP) depends on the chemistry of the components. The overall oxidative stability of the combined solvent-anion system is often significantly lower than the stability of each individual species, and increasing the IP of one of them does not necessarily increase the stability of the resulting electrolyte. By computationally examining a wide range of anion and solvent combinations we find a universal coupling behavior which is explained by the formation of a charge transfer complex upon oxidation, depending on the IP of anions and solvents and their electrostatic interaction. We construct a sim-

ple model based on this understanding that is able to capture the counterintuitive trends observed in DeltaSCF ionization potentials and predicts trends that are consistent with experimental observations. We emphasize that common semi-local density functionals suffer from charge delocalization errors when describing oxidation of representative molecular clusters and are likely to miss the qualitative features and the magnitude of the charge transfer effect that is determined by the electrostatic interaction between local charges resulting from ionization. Using this model, we show how the IP of the pair can be approximated in a simple way, whether in vacuum or in full solvation. We find that hydrogen abstraction from solvent is not the primary reason for the electrolyte’s weakening but is more likely to subsequently occur after a charge-transfer complex is formed upon oxidation. Thus, this is expected to be a likely common second step in the decomposition process. Results presented here provide direct implications and quantitative rules for designing stable battery electrolytes, emphasizing that both solvent and salt anions must be optimized as a whole.

Changed to (pages 26-27):

This study shows that oxidative stability of Li-ion battery electrolytes is governed by non trivial coupling between anion and solvent and requires their coupling to be simulated explicitly. We find that only one molecule, either the solvent or the anion, loses an electron upon oxidation, but the value of the ionization potential (IP) depends on the chemistry of the components. The overall oxidative stability of the combined solvent-anion system is often significantly lower than the stability of each individual species, and increasing the IP of one of them does not necessarily increase the stability of the resulting electrolyte. By computationally examining a wide range of anion and solvent combinations we find a universal coupling behavior which is explained by the formation of a charge transfer complex upon oxidation, depending on the IP of anions and solvents and their electrostatic interaction. We construct a simple model based on this understanding that is able to capture the counterintuitive trends observed in DeltaSCF ionization potentials and predicts trends that are consistent with experimental observations. We emphasize that common semi-local density functionals suffer from charge delocalization errors when describing oxidation of representative molecular clusters and are likely to miss the qualitative features and the magnitude of the charge transfer effect that is determined by the electrostatic interaction between local charges resulting from ionization. Using this model, we show how the IP of the pair can be approximated in a simple way. **We find that the resulting final state of the electron removal may impact the decomposition process, however more investigation is needed to understand the coupled effects of solvent reorganization, molecular relaxation and how much the charge transfer may impact the subsequent decomposition reaction. In vacuum, we show how the dipole formation may facilitate Hydrogen abstraction as a subsequent step in solvent decomposition.** Results presented here provide direct implications and quan-

titative rules for designing stable battery electrolytes, emphasizing that both solvent and salt anions must be optimized as a whole.

Reviewer: Incidentally, if a static dielectric continuum is used throughout, the calculation is not truly "vertical IP" because the outer shell solvent molecules should not have had time to relax (see above too); only the high frequency dielectric constant should have been used. This is another "approximation" that should be discussed in the manuscript.

If these are addressed I don't need to review the paper again.

Author Reply: The reviewer is correct, this is not what we intended to compute. To be consistent with the vertical ionization picture, we redo the PCM calculations following the method outlined in the Gaussian code^{5,6} to compute the implicit solvation effect on the vertical IP. We find that this does not change the results significantly. We also decided to omit the approximation we proposed for the quantitative effect of implicit solvation on the vertical IP and the solvent weakening IP shift δ , as it no longer holds for the corrected calculations.

Original:

So far we explicitly considered pairs of anion and solvent molecules in vacuum. In this section we examine the effect of solvation on the oxidation energetics and the electrostatic interaction between the electrolyte species. Our main finding is that solvation quantitatively changes the electrostatic dipole energy in the presence of solvent (denoted by δ_{\bullet}) primarily due to the dielectric screening effect due to the solvent, and we explain the trends across several solvent-salt combinations again using a simple electrostatic model derived from the above understanding of the charge-transfer complex. First we look at the dependence of the IP on the number of solvent molecules in the explicitly solvated scenario. We find that the average IP increases with the number of solvents (up to five solvent molecules, see figure S5 in supplementary information). This increasing trend is expected from classical electrostatic energy of a charge in a dielectric medium, and can be understood as a polarization effect of the additional solvents, i.e. that anion (negative species) IP increases and solvent (neutral species) IP decreases with the solvent dielectric constant. We note in passing that in order to obtain accurate IP values for explicitly solvated systems, a proper extrapolation to large system size is needed⁷, which requires expensive simulations that lie outside the scope of our investigation. It is also important to note that in all the explicitly solvated computations there is still only one species that is fully oxidized upon removal of charge (whether it is the anion or one of the solvents). We again emphasize that for this to happen it is critical to choose an exchange correlation functional with minimal delocalization errors, such as M06-HF. Therefore, the smallest unit that is needed to study oxidation is the explicit anion-solvent pair, and addition of the full solvation affects the results only through polarization. To analyze the long-range effect of full solvation, and in particular the change in δ_{\bullet} , we employ the PCM implicit solvent model for all the BF_4^- pairs (i.e., BF_4^- solvated with DMSO,

DME, PC or ACN) with dielectric constants of 46.8 for DMSO, 4.2 for DME, 65.5 for PC, and 35.7 for ACN. We find that the solvent is still oxidized, like in vacuum, and the value of the difference δ_{\bullet} between the IP of the solvent and that of the solvent-anion pair is indeed lower but still significant in the PCM-solvated calculations. The values of δ_{\bullet} are reported in table S7 of the supplementary information. To understand the trend, we introduce the effect of solvation into the model of Equation 3, treating it as an effect of classical dielectric continuum on the energy of point charges. The computational finding that δ decreases in solvated systems is expected, because it represents the classical electrostatic energy of the dipole formation, which should decrease with increasing dielectric constant (denoted by ϵ). The detailed derivation of the effect of full solvation on the value of the dipole stabilization energy δ is given in the supplementary information. Following the discussion the supplementary information (see equation S4), the value of the electrostatic stabilization energy at full solvation is well approximated by $\delta_{\bullet} = \delta/\epsilon$. Thus the value of δ_{\bullet} in PCM-solvation for DME is much closer to the vacuum value of than for the other three solvents, consistent with the much lower dielectric constant. In fact we find that the value of δ_{\bullet} can be approximated even without PCM calculations and only using the equation above starting with the value of δ computed in vacuum and the dielectric constant ϵ of the solvent. Table S7 shows the comparison between this approximation and the computed value for δ_{\bullet} . When we compare the values of δ_{\bullet} from PCM calculations to δ/ϵ (where the vacuum value δ is about 2.8 eV) averaged over 5 configurations chosen close to the IP distribution peak, we find that the difference ranges from 0.03 to 0.07 eV. Therefore we have a practical, fast recipe for estimating IP of the full solvent-anion system that requires only computations of IP of individual solvated species.

Changed to (pages 25-26):

So far we explicitly considered pairs of anion and solvent molecules in vacuum. In this section we examine the effect of solvation on the oxidation energetics and the electrostatic interaction between the electrolyte species. Our main finding is that solvation quantitatively changes the electrostatic dipole energy in the presence of solvent (denoted by δ_{\bullet}) primarily due to the dielectric screening effect due to the solvent, and we explain the trends across several solvent-salt combinations again using a simple electrostatic model derived from the above understanding of the charge-transfer complex. First we look at the dependence of the IP on the number of solvent molecules in the explicitly solvated scenario. We find that the average IP increases with the number of solvents (up to five solvent molecules, see figure S5 in supplementary information). This increasing trend is expected from classical electrostatic energy of a charge in a dielectric medium, and can be understood as a polarization effect of the additional solvents, i.e. that anion (negative species) IP increases and solvent (neutral species) IP decreases with the solvent dielectric constant. We note in passing that in order to obtain accurate IP values for explicitly solvated

systems, a proper extrapolation to large system size is needed⁷, which requires expensive simulations that lie outside the scope of our investigation. It is also important to note that in all the explicitly solvated computations there is still only one species that is fully oxidized upon removal of charge (whether it is the anion or one of the solvents). We again emphasize that for this to happen it is critical to choose an exchange correlation functional with minimal delocalization errors, such as M06-HF. Therefore, the smallest unit that is needed to study oxidation is the explicit anion-solvent pair, and addition of the full solvation affects the results only through polarization. To analyze the long-range effect of solvation and estimate δ_{\bullet} , we build on the previous finding with explicit solvents that only one molecule is oxidized and assume that the electron removal is much faster than any other process, obtaining equation S4. We employ the PCM implicit solvent model for all the BF_4^- pairs (i.e., BF_4^- solvated with DMSO, DME, PC or ACN) to estimate δ_{\bullet} . In order to compute the effect of PCM implicit solvation on the vertical IP, one must account only for the optical screening effect of solvation, which we do with the method provided in the Gaussian code^{5,6}. We use the following static and high-frequency dielectric constants ($\epsilon_0, \epsilon_{\infty}$) of (46.8,4.16) for DMSO, (4.24,2.16) for DME, (65.5,4.14) for PC, and (35.7,4.0) for ACN^{8,9}. We find that the solvent is still oxidized, like in vacuum, and the value of the difference δ_{\bullet} between the IP of the solvent and that of the solvent-anion pair is indeed lower but still significant in the PCM-solvated calculations. The values of δ_{\bullet} are reported in table S7 of the supplementary information. Thus we find that implicit solvation does not change the qualitative picture seen in the vacuum case. As discussed above, after the electron removal, it is very difficult to assess how the solvent reorganization and other subsequent reactions are affected by the nature of the oxidized state.

Original Supplementary Material:

We note that these two quantities are positive. Indeed, the classical electrostatic binding energy of a dipole charge-transfer complex $[A^-S^+]$ is expected to be larger in absolute value than that of the unoxidized $[A^-S^0]$ pair. This is verified in the computations highlighted in the article, where we find that for all cases where the solvent is oxidized, the IP of the couple is lower than that of the isolated solvent. As presented in the article, across all couples studied here, the difference in IP is 2.8 eV on average in vacuum, and always positive also in the solvated cases. We also expect that the electrostatic energies (and therefore the binding energies) of the solvated configurations scale inversely with the dielectric constant ϵ of the solvent medium, so that δ_{\bullet} can be expressed as a function of δ :

$$\delta_{\bullet} = E^{\text{bind}}([A^-S^0]_{\bullet}) - E^{\text{bind}}([A^-S^+]_{\bullet}) = \frac{E^{\text{bind}}([A^-S^0]) - E^{\text{bind}}([A^-S^+])}{\epsilon} = \frac{\delta}{\epsilon} \tag{S4}$$

Table S7: Values of δ_{\bullet} for all pairs with BF_4^- anion. δ_{\bullet} is defined as the IP shift that is the difference of the pair IP (in implicit solvation) and the IP of the lone solvent (in implicit solvation) in the case where the solvent is oxidized (column 1). This is averaged over 5 different configurations picked close to the IP distribution peak for the pair (and in the case of ACN, only configurations leading to solvent oxidation). Column 2 represents the value of $\frac{\delta}{\epsilon}$ for the same couples, taking δ to be 2.8 eV and ϵ to be the dielectric constants for the solvents as described above.

Couples	δ_{\bullet}	$\frac{\delta}{\epsilon}$
$\text{BF}_4^- + \text{DMSO}$	0.09	0.05
$\text{BF}_4^- + \text{DME}$	0.61	0.63
$\text{BF}_4^- + \text{PC}$	0.13	0.05
$\text{BF}_4^- + \text{ACN}$	0.16	0.09

For all couples with the BF_4^- anion, we compute the value of δ_{\bullet} with 5 configurations of anion-solvent pairs and using implicit solvation to approximate the full solvation of this pair. The dielectric constants for the solvents are taken to be 46.8 for DMSO, for 4.2 DME, 65.5 for PC and 35.7 for ACN. We find that the solvent is still the oxidized species, just like in the vacuum case, and the value of the difference δ_{\bullet} between the IP of the solvent and that of the solvent-anion pair is indeed lower but still significant in the PCM-solvated calculations. The results are presented in the table S7. We find that we can relatively well approximate δ_{\bullet} by $\frac{\delta}{\epsilon}$, taking the value of δ to be 2.8 eV. The discrepancies are larger for the solvents with lower dielectric constants, (but still on the order of 0.05-0.07 eV). This could be because we have not sampled enough configurations to compute δ_{\bullet} , or because instead of using the specific value of δ for every chemistry, we used the 2.8 eV value which is averaged over all the pairs we studied in this work (although the differences are very small, they are also on the order of the discrepancies we observe here).

Changed to (Supplementary Material, pages 19-20):

We note that these two quantities are positive. Indeed, the classical electrostatic binding energy of a dipole charge-transfer complex $[A^-S^+]$ is expected to be larger in absolute value than that of the unoxidized $[A^-S^0]$ pair. This is verified in the computations highlighted in the article, where we find that for all cases where the solvent is oxidized, the IP of the couple is lower than that of the isolated solvent. As presented in the article, across all couples studied here, the difference in IP is 2.8 eV on average in vacuum, and always positive also in the solvated cases. For all couples with the BF_4^- anion, we compute the value of δ_{\bullet} with 5 configurations of anion-solvent pairs and using implicit solvation to approximate the full solvation of this pair, using the method described in main paper to obtain the compute the change on the vertical IP. The static and high-frequency dielectric constants ($\epsilon_0, \epsilon_{\infty}$) for the solvents are taken to be (46.8,4.16) for DMSO, for (4.24,2.16) DME, (65.5,4.14) for PC and (35.7,4.0) for ACN. We find that the solvent is still the oxidized species, just like in the

Table S7: Values of δ_{\bullet} for all pairs with BF_4^- anion. δ_{\bullet} is defined as the IP drop that is the difference of the pair IP (in implicit solvation) and the IP of the lone solvent (in implicit solvation) in the case where the solvent is oxidized (column 1). This is averaged over 5 different configurations picked close to the IP distribution peak for the pair (and in the case of ACN, only configurations leading to solvent oxidation).

Couples	δ_{\bullet}
$\text{BF}_4^- + \text{DMSO}$	0.1
$\text{BF}_4^- + \text{DME}$	0.64
$\text{BF}_4^- + \text{PC}$	0.12
$\text{BF}_4^- + \text{ACN}$	0.16

vacuum case, and the value of the difference δ_{\bullet} between the IP of the solvent and that of the solvent-anion pair is indeed lower but still significant in the PCM-solvated calculations. The results are presented in the table S7.

References

- [1] Borodin, O.; Jow, T. R. *ECS Transactions* **2011**, *33*, 77–84.
- [2] Borodin, O.; Behl, W.; Jow, T. R. *The Journal of Physical Chemistry C* **2013**, *117*, 8661–8682.
- [3] Kim, D. Y.; Park, M. S.; Lim, Y.; Kang, Y.-S.; Park, J.-H.; Doo, S.-G. *Journal of Power Sources* **2015**, *288*, 393–400.
- [4] Faglioni, F.; Merinov, B. V.; Goddard, W. A.; Kozinsky, B. *Physical Chemistry Chemical Physics* **2018**, *20*, 26098–26104.
- [5] Frisch, M. J.; al., Gaussian 03, Revision C.02. Gaussian, Inc., Wallingford, CT, 2004.
- [6] Tomasi, J.; Mennucci, B.; Cammi, R. *Chemical reviews* **2005**, *105*, 2999–3094.
- [7] Tazhigulov, R. N.; Bravaya, K. B. *J. Phys. Chem. Lett* **2016**, *7*, 2490–2495.
- [8] Barthel, J.; Bachhuber, K.; Buchner, R.; Hetzenauer, H. *Chemical physics letters* **1990**, *165*, 369–373.
- [9] Cachet, H.; Fekir, M.; Lestrade, J.-C. *Canadian Journal of Chemistry* **1981**, *59*, 1051–1060.